# On the approximation properties of recurrent encoder-decoder architectures

**Zhong Li**[*]
School of Mathematical Sciences
Peking University
li_zhong@pku.edu.cn

**Haotian Jiang**[*]
Department of Mathematics
National University of Singapore
e0012663@u.nus.edu

**Qianxiao Li**[†]
Department of Mathematics
National University of Singapore
qianxiao@nus.edu.sg

## Abstract

Encoder-decoder architectures have recently gained popularity in sequence to sequence modelling, featuring in state-of-the-art models such as transformers. However, a mathematical understanding of their working principles still remains limited. In this paper, we study the approximation properties of recurrent encoder-decoder architectures. Prior work established theoretical results for RNNs in the linear setting, where approximation capabilities can be related to smoothness and memory of target temporal relationships. Here, we uncover that the encoder and decoder together form a particular "temporal product structure" which determines the approximation efficiency. Moreover, the encoder-decoder architecture generalises RNNs with the capability to learn time-inhomogeneous relationships. Our results provide the theoretical understanding of approximation properties of the recurrent encoder-decoder architecture, which precisely characterises, in the considered setting, the types of temporal relationships that can be efficiently learned.

## 1 Introduction

Encoder-decoder is an increasingly popular architecture for sequence to sequence modelling problems (Sutskever et al., 2014; Chiu et al., 2018; Venugopalan et al., 2015). The core of this architecture is to first encode the input sequence into a vector using the encoder and then map the vector into the output sequence through the decoder. In particular, such architecture forms the main component in the transformer network (Vaswani et al., 2017), which has become a powerful method for modelling sequence to sequence relationships (Parmar et al., 2018; Beltagy et al., 2020; Li et al., 2019).

The encoder-decoder family of structures differ significantly from direct application of recurrent neural networks (RNNs, Elman (1990)) and its generalisations (Hochreiter & Schmidhuber, 1997; Cho et al., 2014b) for processing sequences. However, both architectures can be considered as modelling mappings between sequences, albeit with different underlying structures. Hence, a natural but unresolved question is: how are these approaches fundamentally different? Answering this question is not only of theoretical importance but also of practical interest. Currently, architectural selection for different time series modelling tasks is predominantly empirical. Thus, it is desirable to develop a concrete mathematical framework to understand the key differences between separate architectures in order to guide practitioners in a principled way.

---

[*]Equal contribution
[†]Corresponding author

In this paper, we investigate the approximation properties of encoder-decoder architectures. Approximation is one of the most basic and important problems for supervised learning. It considers to what extent the model can fit a target. In particular, we prove a general approximation result in the linear setting, which characterises the types of temporal input-output relationships that can be efficiently approximated by encoder-decoder architectures. These results reveal that such architectures essentially generalise RNNs by lifting the requirement of time-homogeneity (see Remark 3.2) in the target relationships. Hence, it can be used to tackle a broader class of sequence to sequence problems. Furthermore, of particular interest is the identification of a "temporal product structure" — a precise property of the target temporal relationship that highlights another intrinsic difference between recurrent encoder-decoders and RNNs.

Our main contributions can be summarised as follows.

1. We prove a universal approximation result for recurrent encoder-decoder architectures in the linear setting, including the approximation rates.

2. We show that in the considered setting, the recurrent encoder-decoder generalises the RNNs and can approximate time-inhomogeneous relationships, which further adapt to additional temporal product structures in the target relationship. This answers precisely how encoder-decoders are different from RNNs, at least in the considered setting.

**Organisation.** In Section 2, we review the related work on encoder-decoder architectures and general approximation theories of sequence modelling. The approximation problem is formulated in Section 3. Our main results, their consequences and numerical illustrations are presented in Section 4. All the proofs and numerical details are included in appendices.

**Notations.** For consistency, we adhere to the following notations. Boldfaced letters are reserved for sequences or paths, which can be understood as functions of time. Lower case letters can mean vectors or scalars. Matrices are denoted by capital letters. For $\alpha \in \mathbb{N}$, $C^\alpha$ denotes the space of functions with continuous derivatives up to order-$\alpha$.

## 2 RELATED WORK

We first review some previous works on sequence to sequence modelling. The encoder-decoder architecture first appeared in Kalchbrenner & Blunsom (2013), where they map the input sequence into a vector using convolutional neural networks (CNNs), and then using a recurrent structure to map the vector to the output sequence. With the flexibility of manipulating the underlying structure of encoder and decoder, numerous models based on this architecture have come out thereafter. For instance, Cho et al. (2014b) used gated RNNs as both the encoder and decoder, while in the later work (Cho et al., 2014a), they proposed a CNN-based decoder. In Sutskever et al. (2014), they proposed a deep LSTM for both the encoder and decoder. Bahdanau et al. (2015) first introduced the attention mechanism, which was further developed in the well-known transformer networks (Vaswani et al., 2017). However, most of the research on encoder-decoder architectures focused on applications. A theoretical understanding is helpful for its further improvement and development.

From the theoretical point of view, Ye & Sung (2019) studied several theoretical properties of CNN encoder-decoders, including expressiveness, generalisation capability and optimisation landscape. Of particular relevance to the current work is expressiveness, which considers the relationships that can be generated from the architecture. However, this is not approximation. Yun et al. (2020) proved the universal approximation property of transformers for certain classes of functions, for example, permutation equivariant functions, but they did not consider the actual dynamical properties of target relationships that affect approximation. Dynamical proprieties such as memory, smoothness and low rank structures are essential, because they can precisely characterise different temporal relationships and affect the approximation capabilities of models. Assuming the target generated from a hidden dynamical system is one approach, which is widely applied (Maass et al., 2007; Schäfer & Zimmermann, 2007; Doya, 1993; Funahashi & Nakamura, 1993). In contrast, a functional-based approach is

recently introduced, where the target temporal relationships are generated from functionals satisfying specific properties such as linearity, continuity, regularity and time-homogeneity (Li et al., 2021). In Li et al. (2021), the approximation properties of linear RNN models are studied, and the results therein show that the approximation efficiency is related to the memory structure. In Jiang et al. (2021), similar formulations are applied to investigate convolutional architectures, where the results suggest that targets with certain spectrum regularity can be well approximated by dilated CNNs. Under this framework, the target temporal relationship that can be efficiently approximated is characterised by properties such as memory, smoothness and sparsity. This enables us to make precise mathematical comparisons between different architectures. Our results in this work reveal that the encoder-decoders have a special temporal product structure which is intrinsically different from other sequence modelling architectures.

## 3 PROBLEM FORMULATION

In this section, we precisely define the input space, output space, concept space and hypothesis space, respectively.

**Functional formulation of temporal modelling.** First, we define the input and output space precisely. A temporal sequence can be viewed as a function of time $t$. The input space is defined by $\mathcal{X} = C_0((-\infty, 0], \mathbb{R}^d)$. This is the space of continuous functions from $(-\infty, 0]$ to $\mathbb{R}^d$ vanishing at infinity, where $d \in \mathbb{N}_+$ is the dimension. Denote the element in $\mathcal{X}$ by $\boldsymbol{x} = \{x_t \in \mathbb{R}^d : t \leq 0\}$, we equip $\mathcal{X}$ with the supremum norm $\|\boldsymbol{x}\|_{\mathcal{X}} := \sup_{t \leq 0} \|x_t\|_\infty$. We take the outputs space as $\mathcal{Y} = C_b([0, \infty), \mathbb{R})$, the space of bounded continuous functions from $[0, \infty)$ to $\mathbb{R}$. We consider real-valued outputs, since each dimension can be handled individually for vector-valued outputs.

The mapping between input and output sequences can be formulated as a *sequence of functionals*, i.e. $y_t = H_t(\boldsymbol{x})$, $t \geq 0$. The output $y_t$ at the time step $t$ depends on the input sequence $\boldsymbol{x}$. The ground truth relation between inputs and outputs is formulated by the sequence of functionals $\boldsymbol{H} := \{H_t : t \geq 0\}$.

We provide an example to illustrate the above formulation. Given an input $\boldsymbol{x}$, the output $\boldsymbol{y}$ is a smoothed version of $\boldsymbol{x}$, resulting from convolving $\boldsymbol{x}$ with the Gaussian kernel $g(s) = \frac{1}{\sqrt{2\pi}} \exp(-\frac{s^2}{2})$. This relation can be formulated as $y_t = H_t(\boldsymbol{x}) = \int_0^\infty g(t + s)\boldsymbol{x}_{-s}ds$.

**The RNN encoder-decoder model.** For the supervised learning problem, our goal is to use a model to learn the target relationship $\boldsymbol{H}$. First, we define the model. Among all different variants of the encoder-decoder architectures, the RNN encoder-decoder introduced in Cho et al. (2014b) can be considered as the most simple and representative model, where the encoder and decoder are both RNNs. We study this particular model as we try to eliminate other factors and only focus on the encoder-decoder architecture itself.

Under our setting, the simplified model of Cho et al. (2014b) with RNNs as both encoder and decoder can be formulated as

$$
\begin{aligned}
h_s &= \sigma_E(W_E h_{s-1} + U_E x_s + b_E), & v &= h_\tau, \\
g_t &= \sigma_D(W_D g_{t-1} + b_D), & g_0 &= v, \\
o_t &= W_O g_t + b_O,
\end{aligned}
\tag{1}
$$

where $h_t$, $g_t$ are *hidden states* of the encoder and decoder respectively. Recurrent activation functions are denoted by $\sigma_E$ and $\sigma_D$. Here, $\tau$ denotes the terminating time step of the encoder, and $v$ is the summary of the input sequence, which is called as the coding vector. The model prediction is denoted as $o_t \in \mathbb{R}$. All the other notations are model parameters. Equation (1) describes the following model dynamics. First, the encoder reads the entire input $\boldsymbol{x}$, and then summarises the input into a fixed size coding vector $v$, which is also the last hidden state of the encoder. Next, the coding vector is passed into the decoder as the initial state, and then the decoder produces an output at each time step. Note that the encoder has a terminating time, and the decoder has a starting time. This is the reason why we take the input and output as semi-infinite sequences.

We study a linear, residual and continuous-time idealisation of the model dynamics (1):

$$
\begin{aligned}
\dot{h}_s &= W h_s + U x_s, & v &= Q h_0, & s &\leq 0 \\
\dot{g}_t &= V g_t, & g_0 &= P v, & & \\
o_t &= c^\top g_t, & t &\geq 0, & &
\end{aligned}
\tag{2}
$$

where $W \in \mathbb{R}^{m_E \times m_E}$, $U \in \mathbb{R}^{m_E \times d}$, $Q \in \mathbb{R}^{N \times m_E}$, $V \in \mathbb{R}^{m_D \times m_D}$, $P \in \mathbb{R}^{m_D \times N}$ and $c \in \mathbb{R}^{m_D}$ are parameters. $m_E$ and $m_D$ denote the width of encoder and decoder, respectively. The coding vector $v$ has dimension $N$, where we apply linear transformations to control it. We assume $h_{-\infty} = 0$, which is the usual choice for the initial condition of RNN hidden states.

Since our goal is to investigate approximation problems over large time horizons, we are supposed to consider the stable RNN encoder-decoders, where

$$
W \in \mathcal{W}_{m_E} := \{ W \in \mathbb{R}^{m_E \times m_E} : \text{eigenvalues of } W \text{ have negative real parts} \}, \tag{3}
$$

$$
V \in \mathcal{V}_{m_D} := \{ V \in \mathbb{R}^{m_D \times m_D} : \text{eigenvalues of } V \text{ have negative real parts} \}. \tag{4}
$$

The hypothesis space of RNN encoder-decoder models with arbitrary widths and coding vector dimension is defined as $\widehat{\mathcal{H}} := \bigcup_{m_E, m_D, N \in \mathbb{N}_+} \widehat{\mathcal{H}}_{m_E, m_D, N}$, where

$$
\begin{aligned}
\widehat{\mathcal{H}}_{m_E, m_D, N} := \Big\{ & \widehat{\boldsymbol{H}} := \{ \widehat{H}_t : t \geq 0 \} : \widehat{H}_t(\boldsymbol{x}) = c^\top e^{Vt} P \int_0^\infty Q e^{Ws} U x_{-s} ds, \text{ with} \\
& (W, U, Q, V, P, c) \in \mathcal{W}_{m_E} \times \mathbb{R}^{m_E \times d} \times \mathbb{R}^{N \times m_E} \times \mathcal{V}_{m_D} \times \mathbb{R}^{m_D \times N} \times \mathbb{R}^{m_D} \Big\}.
\end{aligned}
\tag{5}
$$

The widths $m_E, m_D$ and the coding vector dimension $N$ together control the capacity/complexity of the hypothesis space. Note that the assumptions on eigenvalues of $W$ and $V$ ensure that the parameterized linear functionals are continuous.

Due to the mathematical form (5), not all functionals can be represented by RNN encoder-decoders. To achieve a good approximation, the target functionals must possess certain structures. We introduce the following definitions to clarify these structures.

**Definition 3.1.** *Let $\boldsymbol{H} = \{ H_t : t \geq 0 \}$ be a sequence of functionals.*

1. *For any $t \geq 0$, the functional $H_t$ is* linear *and* continuous *if for any $\lambda_1, \lambda_2 \in \mathbb{R}$ and $\boldsymbol{x}_1, \boldsymbol{x}_2 \in \mathcal{X}$, we have $H_t(\lambda_1 \boldsymbol{x}_1 + \lambda_2 \boldsymbol{x}_2) = \lambda_1 H_t(\boldsymbol{x}_1) + \lambda_2 H_t(\boldsymbol{x}_2)$, and $\| H_t \| := \sup_{\boldsymbol{x} \in \mathcal{X}, \| \boldsymbol{x} \|_{\mathcal{X}} \leq 1} | H_t(\boldsymbol{x}) | < \infty$, where $\| H_t \|$ denotes the induced functional norm.*

2. *For any $t \geq 0$, the functional $H_t$ is* regular *if for any sequence $\{ \boldsymbol{x}^{(n)} \}_{n=1}^\infty \subset \mathcal{X}$ such that $\lim_{n \to \infty} x_s^{(n)} = 0$ for almost every $s \leq 0$ (Lebesgue measure), we have $\lim_{n \to \infty} H_t(\boldsymbol{x}^{(n)}) = 0$.*

*For a sequence of functionals $\boldsymbol{H}$, we define its norm by $\| \boldsymbol{H} \| := \int_0^\infty \| H_t \| dt$.*

**Remark 3.1.** *The definitions of linear and continuous functionals are standard. One can view regular functionals as those not determined by inputs on arbitrarily small time intervals, e.g. an infinitely thin spike (i.e. $\delta$-functions).*

Given the above definitions, we immediately have the following observation.

**Proposition 3.1.** *Let $\widehat{\boldsymbol{H}} \in \widehat{\mathcal{H}}$ be a sequence of functionals in the RNN encoder-decoder hypothesis space (see (5)). Then for any $t \geq 0$, $\widehat{H}_t \in \widehat{\boldsymbol{H}}$ is a linear, continuous and regular functional. Furthermore, $\| \widehat{H}_t \|$ decays exponentially as a function of $t$.*

The proof is found in Appendix A. This proposition characterises properties of the encoder-decoder hypothesis space. In particular, it is different from the RNN hypothesis space discussed in Li et al. (2021), since the encoder-decoder is not necessarily *time-homogeneous*.

**Remark 3.2.** *A sequence of functionals $\boldsymbol{H}$ is time-homogeneous if for any $t, \tau \geq 0$, $H_t(\boldsymbol{x}) = H_{t+\tau}(\boldsymbol{x}(\tau))$, with $x(\tau)_s = x_{s-\tau}$ for all $s \in \mathbb{R}$. That is, if the input is shifted to the right by*

$\tau$, the output is also shifted by $\tau$. Temporal convolution is an example of time-homogeneous operation (recall the Gaussian convolution discussed in Section 3. An example of time-inhomogeneous relationship is video captioning: shifts in the sequence of input video frames do not necessarily lead to corresponding shifts in the caption text sequence.

**Relation with RNNs.** Here, we emphasise the differences between the encoder-decoder hypothesis space and the RNN hypothesis space discussed in Li et al. (2021), where $\widehat{H}_t^{(\mathrm{RNN})}(\boldsymbol{x}) = \int_0^\infty c^\top e^{W(t+s)} U x_{-s} ds$. A key difference is that the encoder-decoder has a structure involving two temporal parameters $t$ and $s$, while the RNN only has one depending on $t+s$, due to the time-homogeneity. Owing to this difference and the fact that $\widehat{\mathcal{H}}^{(\mathrm{RNN})} \subset \widehat{\mathcal{H}}$, the encoder-decoder hypothesis space (5) is more general, with the extra capability to learn time-inhomogeneous relationships. Furthermore, $e^{Vt}$ and $e^{Ws}$ adapt to a temporal product structure, which is an intrinsic difference between encoder-decoders and other architectures. We will discuss this in detail in the next section.

## 4 APPROXIMATION RESULTS

One of the most fundamental problems for supervised learning is the approximation problem. It basically concerns the capacity of the hypothesis space to fit the concept space. In general, there are two levels of approximation problems that can be discussed. The first is known as the universal approximation, which considers the density of the hypothesis space in the concept space. The second is the approximation rate, which aims to characterise quantitatively the approximation accuracy concerning the capacity/complexity of the hypothesis space (e.g. the number of trainable parameters). In this section, both of them are developed for RNN encoder-decoders.

### 4.1 UNIVERSAL APPROXIMATION

We first present the most basic density result, which states that any linear, continuous, and regular temporal relationship can be approximated by RNN encoder-decoders up to arbitrary accuracy. The proof is found in Appendix B.

**Theorem 4.1.** *Let $\boldsymbol{H}$ be a sequence of linear, continuous and regular functionals defined on $\mathcal{X}$, and satisfy $\|\boldsymbol{H}\| < \infty$. Then for any $\epsilon > 0$, there exists $\widehat{\boldsymbol{H}} \in \widehat{\mathcal{H}}$ such that*

$$\|\boldsymbol{H} - \widehat{\boldsymbol{H}}\| \equiv \int_0^\infty \|H_t - \widehat{H}_t\| dt < \epsilon. \tag{6}$$

Here, we highlight two important observations while deriving Theorem 4.1. First, one can show that each sequence of functionals $\boldsymbol{H} \in \mathcal{H}$ can be associated with a unique two-parameter "representation" $\rho(t,s)$, such that $H_t(\boldsymbol{x}) = \int_0^\infty x_{-s}^\top \rho(t,s) ds$. Recall the model form $\widehat{H}_t(\boldsymbol{x}) = \int_0^\infty x_{-s}^\top \hat{\rho}(t,s) ds$, where $\hat{\rho}(t,s) := [c^\top e^{Vt} PQ e^{Ws} U]^\top$ denotes the corresponding representation. The functional approximation is then reduced to function approximation in the sense of representations, i.e. $\|\boldsymbol{H} - \widehat{\boldsymbol{H}}\| \leq \|\rho - \hat{\rho}\|_{L^1([0,\infty)^2)}$. It turns out that $\rho$ directly affects the rate of approximation and gives rise to intrinsic properties. We will discuss this in detail in Section 4.3. In addition, we again emphasise the differences between the present work and Li et al. (2021). In Li et al. (2021), the target relationships are assumed to be time-homogeneous with the representation $H_t(\boldsymbol{x}) = \int_0^\infty \rho(t+s) x_{-s} ds$, which only depends on $t+s$. However, the setting here does not assume time-homogeneity, hence implies a more general representation $\rho$ depending on the two temporal directions $t$ and $s$ simultaneously.

### 4.2 GENERAL APPROXIMATION RATES

While the density result (Theorem 4.1) ensures the universal approximation property of the RNN encoder-decoder, it does not identify targets that can be efficiently approximated. To achieve this, we focus on approximation rates next. We characterise the temporal structure of a target relationship by observing its responses to "constant" input signals. Here, we consider the approximation rates for the model with "large size" coding vector, where the

dimension $N \geq \bar{m} := \min\{m_E, m_D\}$. This is the scenario where we fix the widths but take an oversized coding vector.

**Theorem 4.2.** *Let $\boldsymbol{H}$ be a sequence of linear, continuous and regular functionals defined on $\mathcal{X}$, and satisfy $\|\boldsymbol{H}\| < \infty$. Consider the output of piece-wise constant signals $y_i^c(t, s) = H_t(e_i \mathbf{1}_{(-\infty, -s]})$, $t, s \geq 0$, $i = 1, 2, \ldots, d$, where $\{e_i\}_{i=1}^d$ denotes the standard basis of $\mathbb{R}^d$. Assume that there exist $\alpha \in \mathbb{N}_+$, $\beta > 0$ such that for any $i = 1, 2, \ldots, d$,*

$$y_i^c \in C^{\alpha+1}([0, \infty)^2), \tag{7}$$

$$e^{\beta(t+s)} \frac{\partial^{k+l}}{\partial t^k \partial s^l} y_i^c(t, s) = o(1) \text{ as } \|(t, s)\| \to \infty, \quad (k, l) \in \mathbb{N} \times \mathbb{N}_+, \; k + l \leq \alpha + 1. \tag{8}$$

*Then for any $m_E, m_D, N \in \mathbb{N}_+$, there exists $\widehat{\boldsymbol{H}} \in \widehat{\mathcal{H}}_{m_E, m_D, N}$ such that*

$$\|\boldsymbol{H} - \widehat{\boldsymbol{H}}\| \leq \frac{C(\alpha)\gamma d}{\beta^2}\left(\frac{1}{m_E^\alpha} + \frac{1}{m_D^\alpha}\right), \tag{9}$$

*where $C(\alpha), \gamma > 0$ are both universal constants with dependence only on $\alpha$ and $(\alpha, \beta)$, respectively, and $\gamma := \max\limits_{i \in \mathbb{N}_+, \, i \leq d} \max\limits_{k, l \in \mathbb{N}, \, k+l \leq \alpha+1} \sup\limits_{t, s \geq 0} \beta^{-(k+l)} e^{\beta(t+s)} \left|\frac{\partial^{k+l}}{\partial t^k \partial s^l} y_i^c(t, s)\right| < \infty$. Here, the number of trainable parameters is $dN(m_E + m_D)$ with $N \geq \bar{m}$.*

The proof is found in Appendix C. First, note that the error bound does not depend on the coding vector size $N$, as long as $N \geq \bar{m}$. This is because further increasing $N$ beyond $\bar{m}$ only increases the number of trainable parameters, but does not increase the model capacity (see Remark C.2). Only the model widths $m_E, m_D$ affect the approximation capabilities.

Next, we focus on the classes of target relationships that can be well approximated. Here, $\alpha$ characterises the smoothness of $\boldsymbol{H}$, and $\beta$ characterises the temporal decay rates of the output responding to a constant signal under $\boldsymbol{H}$. This is a notion of memory in the target relationship. The error bound (9) indicates that a sequence of target functionals can be efficiently approximated by the encoder-decoder if it is smooth (large $\alpha$), and has fast decayed memory (large $\beta$).

The characterisation in smoothness and memory decay also appears in the approximation results of RNNs (Li et al., 2021), where the upper bound is $\frac{C(\alpha)\gamma d}{\beta m^\alpha}$. However, our results for encoder-decoders suggest extra structures, where the bound involves two (instead of one) temporal parameters together with smoothness and decay memories in both. The two-parameter temporal dependence allows the encoder-decoder to approximate time-inhomogeneous relationships, which generalises the RNN. This two-parameter structure further leads to adaptation to a specific low rank type of target relationships, resulting in finer approximation rates as we discuss next.

### 4.3 APPROXIMATION RATES VIA TEMPORAL PRODUCT STRUCTURE

**Motivation of temporal product structure.** In contrast with Theorem 4.2, we next consider the model with $N < \bar{m} = \min\{m_E, m_D\}$. In this situation, the model has fewer parameters, and we aim to characterise the target relationships by further exploiting the structure of the two-parameter representation $\rho(t, s)$. This leads to a finer approximation rate by considering $m_E, m_D, N$ together.

We first motivate how the "temporal product structure" arises, and how it relates to the approximation. Detailed discussions and proofs are found in Appendix D. For the illustration purpose, we set the input dimension $d = 1$. Recall $Q \in \mathbb{R}^{N \times m_E}$, $P \in \mathbb{R}^{m_D \times N}$, then the representation $\hat{\rho}$ of the encoder-decoder functional can be rewritten as

$$\hat{\rho}(t, s) = c^\top e^{Vt} P \cdot Q e^{Ws} u = \sum_{n=1}^N \left(\sum_{i,j=1}^{m_D} c_i P_{jn} \left[e^{Vt}\right]_{ij}\right)\left(\sum_{i,j=1}^{m_E} u_i Q_{nj} \left[e^{Ws}\right]_{ji}\right)$$

$$= \sum_{n=1}^N \hat{\varphi}_n(t)\hat{\phi}_n(s). \tag{10}$$

This is a tensor product structure over the $(t, s)$ time domain (determined by the encoder $\{\hat{\phi}_n\}$ and decoder $\{\hat{\varphi}_n\}$ successively). We call it the *temporal product structure*. As is shown later, this structure significantly affects approximation rates. When $\{\hat{\phi}_n\}$ and $\{\hat{\varphi}_n\}$ are selected as the "bases" along $s, t$ direction, respectively, $N$ is considered as the *rank* of the temporal product. We also define $N$ as the rank of the model, which is understood as the maximum rank of temporal products that the encoder-decoder model can represent.

**The rank concept of temporal relationships.** Recall that the given number of trainable parameters is $dN(m_E + m_D)$. Hence, a low rank model may achieve fewer trainable parameters. When investigating relationships that can be well approximated by low rank models, a natural conjecture would be "low rank" targets.

What is the meaning of "low rank" for a temporal relationship? It is well-known that in linear algebra, an operator is low rank means that its range space is low-dimensional. This idea can be also applied to temporal relationships. For a "low rank" temporal relationship, the output sequence is more "regular", meaning that the output sequences (viewed as functions) are in a low-dimensional function space. We provide an intuitive numerical illustration for better understanding.

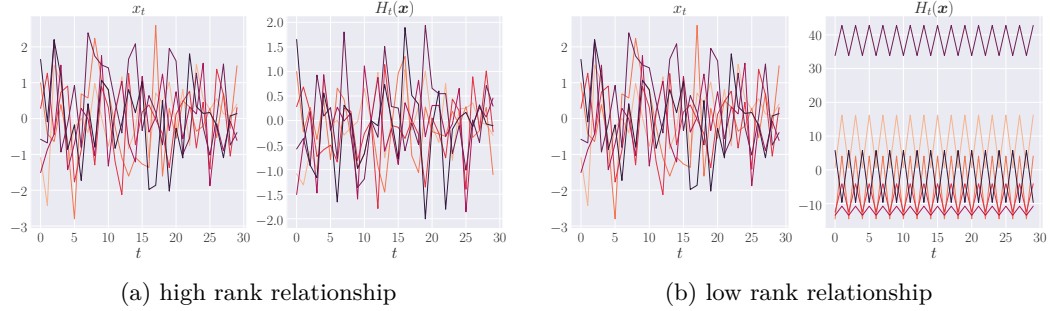

(a) high rank relationship          (b) low rank relationship

Figure 1: We construct a high rank and a low rank target from the temporal product. For both (1a) and (1b), we plot the inputs $x_t$ together with the corresponding outputs $H_t(\boldsymbol{x})$. Detailed settings are found in Appendix E.1.

Figure 1 shows the outputs of a high rank (a) and a low rank (b) target relationship on the same set of random input sequences. Different colours refer to different instances of inputs. In the first case (high rank), the temporal structure of the outputs is very complex and depends sensitively on the inputs. However, in the second case (low rank), the output sequences are much more regular, and only macroscopic structures (e.g. scale/offset) appear.

**Remark 4.1.** *In the research of approximation theories for temporal sequences, prior works also related a notion of rank to approximation properties of the dilated convolutional structure (Jiang et al., 2021). Here, we emphasise that the notion of rank considered in our work is very different from that in Jiang et al. (2021), which mainly concerns the tensorisation of a discrete-time sequence according to the width of convolution filters.*

**POD as an analogue of SVD.** Now, we characterise low rank and high rank temporal relationships in a mathematical way. We will introduce the concepts informally, and rigorous definitions and arguments can be found in Appendix D.

For a matrix, we can assess its rank by performing the singular value decomposition (SVD). This method can be extended to the temporal relationships using proper orthogonal decomposition (POD; Liang et al. (2002), Berkooz et al. (1993), Chatterjee (2000)). The basic insight is that the function $\rho$ can be decomposed into the following form: $\rho(t, s) = \sum_{n=1}^{N_0} \sigma_n \varphi_n(t) \phi_n(s)$, where $N_0 \leq \infty$, $\{\varphi_n\}$ and $\{\phi_n\}$ are orthonormal bases, and $\sigma_1 \geq \sigma_2 \geq \cdots \geq 0$ denote the singular values. This procedure can be viewed as applying SVD to an infinite-dimensional space (when $N_0 = \infty$). An analogue of Eckart–Young theorem (Eckart & Young, 1936), which characterises the best low rank approximation, also holds for POD. It roughly states that $\inf_{\text{rank}(\hat{\rho})=N} \|\rho - \hat{\rho}\|_{L^2}^2 = \sum_{n=N+1}^{N_0} \sigma_n^2$. That is,

any target $\rho$ has a rank-$N$ best approximation, with error equalling to the tail sum of the squared singular values. In other words, a target with fast decayed $\sigma_n$ (low "effective rank") has smaller approximation errors. This forms the basis of our next result, which states that if the target relationship possesses an effective low rank structure in terms of the decay of singular values, then one can achieve an efficient approximation using encoder-decoder structures by limiting the size of coding vectors. Detailed definitions for $\{\sigma_n\}$ and proofs are found in Appendix D.

**Theorem 4.3.** *Assume the same conditions as in Theorem 4.2. Then for any $m_E, m_D, N \in \mathbb{N}_+$ with $N \leq \bar{m}$, there exists $\widehat{\boldsymbol{H}} \in \widehat{\mathcal{H}}_{m_E,m_D,N}$ such that*

$$
\|\boldsymbol{H} - \widehat{\boldsymbol{H}}\| \lesssim \frac{C(\alpha)\gamma d}{\beta^2} \Bigg\{ \left(1 + \sqrt{\bar{m} - N}\right) \cdot \left(\frac{1}{m_E^\alpha} + \frac{1}{m_D^\alpha}\right) + \left(\sum_{n=N+1}^{\bar{m}} \sigma_n^2\right)^{1/2}
$$
$$
+ \left(\sum_{n=N+1}^{\bar{m}} \sigma_n\right)^{1/2} \cdot \left(\frac{1}{m_E^{\alpha/2}} + \frac{1}{m_D^{\alpha/2}}\right) \Bigg\}, \tag{11}
$$

*where $\lesssim$ hides universal positive constants, and $\bar{m} = \min\{m_E, m_D\}$. Here, the number of trainable parameters is $dN(m_E + m_D)$ with $N \leq \bar{m}$.*

This is a finer approximation rate compared to Theorem 4.2, where both the widths $m_E, m_D$ and the coding vector size $N$ affect the model capacity for approximation. Besides the smoothness and memory decay, we have the additional rank structure of the target relationship, which is characterised by its singular values $\{\sigma_n\}$. We again focus on the class of functionals that can be well approximated. Smoothness $\alpha$ and decay rate $\beta$ is the same as Theorem 4.2. The difference lies in the rank structure indicated by $\{\sigma_n\}$: the error bound is small if $\{\sigma_n\}$ has a small tail $\sum_{n=N+1}^{\bar{m}} \sigma_n^2$. It suggests that a target with fast decayed $\{\sigma_n\}$ or low "effective rank" can be well approximated by the RNN encoder-decoder with fewer parameters. Due to the Eckart–Young-like low rank approximation, we can appropriately select $N$ based on the decay rate of singular values.

Here, we emphasise that the temporal product is an *intrinsic* structure arising from the encoder-decoder architecture. Recall the dynamics of the encoder-decoder: it first encodes the input sequence into a coding vector, and then decodes an entire output sequence from it. In this sense, the coding vector is the only interaction between the input and output. Thus, the coding vector size $N$ is an essential measure of the model capacity concerning the dependence of outputs on inputs. Here, we show that this concept can be formalised as a notion of rank, which can pinpoint the precise types of input-output relationships that encoder-decoder architectures are well adapted to.

**Numerical illustrations.** Here, we utilise numerical examples to illustrate the above discussions. We observe how the decay rate of singular values, the rank $N_0$ of the target relationships, and the model rank $N$ affect the approximation error $\|\boldsymbol{H} - \widehat{\boldsymbol{H}}\|$. However, it is not always possible to construct the best approximation. Instead, we perform some training steps to achieve an upper bound of the approximation error, which is consistent with our theoretical results.

In Figure 2, we train linear encoder-decoder models to learn three relationships of different ranks determined by various decay patterns of singular values, given in (a), (b) and (c). Different colours denote targets with different ranks. From Figure 2, we have the following observations consistent with previous discussions. First, observe that increasing the model rank $N$ makes approximation errors smaller, as expected. Moreover, note that when increasing $N$, the speeds of error decrements are different. If the singular values decay fast, the approximation errors also decay fast. This implies that a target with fast decayed singular values can be approximated efficiently with fewer parameters (smaller $N$). In addition, for each experiment, we are able to achieve low approximation errors by choosing $N \ll m$. The errors will remain unchanged or decrease much more slowly when further increasing $N$. This suggests that in practice, one can choose $N$ such that it covers the major singular values of the target in order to improve the approximation efficiency.

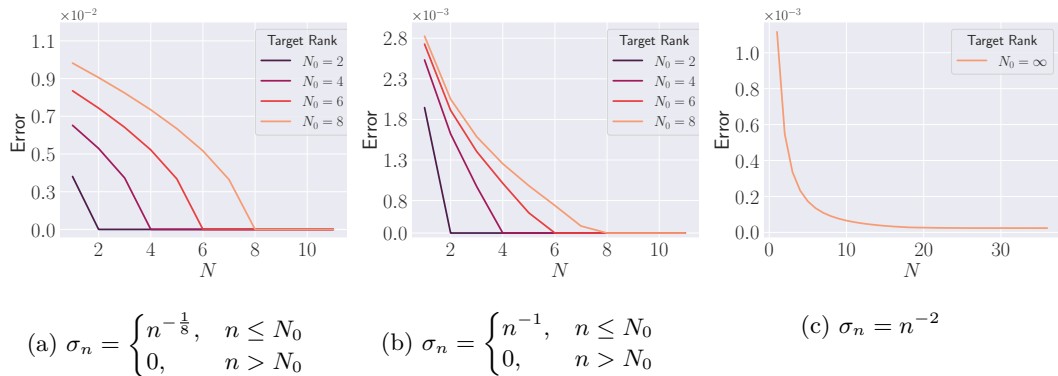

(a) $\sigma_n = \begin{cases} n^{-\frac{1}{8}}, & n \le N_0 \\ 0, & n > N_0 \end{cases}$
(b) $\sigma_n = \begin{cases} n^{-1}, & n \le N_0 \\ 0, & n > N_0 \end{cases}$
(c) $\sigma_n = n^{-2}$

Figure 2: In (a), (b), (c) we consider target relationships with different singular values indicated in the respective caption. For (a), (b) we also consider targets with different rank, where $N_0 = 2, 4, 6, 8$. We use models with fixed width $m = m_E = m_D = 128$ and coding vector size $N$. Detailed settings are found in Appendix E.2.

In Figure 3, we perform experiments on the forced Lorentz 96 system (Lorenz, 1996), which parameterises a high-dimensional and nonlinear relationship between input forcing and model states. The parameters $K, J$ in the Lorenz 96 system control the overall complexity of the target (see Appendix E.3 for details). We use the RNN encoder-decoder with tanh activations to learn this target. Although our theories are developed in the linear regime, the low rank approximation phenomenon also appears in this nonlinear setting. The error decrements saturate when increasing the coding vector size $N$ beyond a threshold, suggesting the existence of some implicit notion of "rank" of the target nonlinear functional. This "rank" increases with the target complexity (mainly $K$).

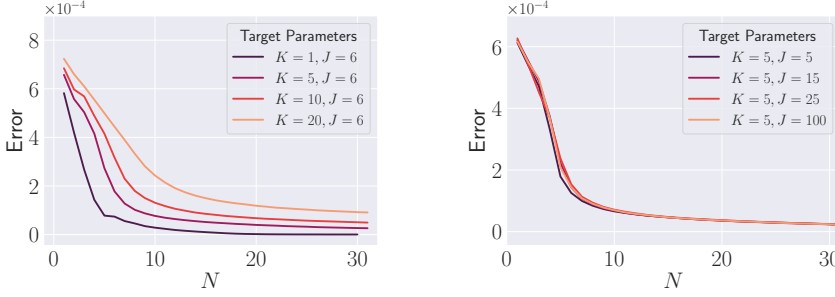

Figure 3: $K, J$ are the parameters of the Lorenz 96 system. They describe the number of independent and coupled variables in the system, which can be viewed as a complexity measure. Detailed settings are found in Appendix E.3.

## 5 CONCLUSION

We theoretically study the approximation properties of the RNN encoder-decoder in a linear setting. We prove a universal approximation result for linear temporal relationships utilising encoder-decoder architectures, and show that they generalise RNNs to the time-inhomogeneous setting. Moreover, we discover an important temporal product structure that characterises the types of input-output relationships especially suited for the efficient approximation using encoder-decoders. This elucidates the key differences between these novel architectures and classical methods for temporal modelling, and forms a basic step towards understanding the intricacies of modern deep learning.

**Reproducibility statements.**   Detailed proofs for theoretical results, and complete settings of numerical examples are found in the appendix. The source code for numerical tests can be made available upon request.

Here is a quick reference:

| | | |
|---|---|---|
| Proposition 3.1 | Properties of RNN encoder-decoder functionals | Appendix A |
| Theorem 4.1 | Universal approximation theorem | Appendix B |
| Theorem 4.2 | General approximation rates | Appendix C |
| Theorem 4.3 | Approximation rates concerning temporal product structure | Appendix D |
| Figure 1 | Illustration of high/low rank temporal relationships | Appendix E.1 |
| Figure 2 | Numerical examples on singular values | Appendix E.2 |
| Figure 3 | Numerical examples on Lorenz 96 systems | Appendix E.3 |

## ACKNOWLEDGEMENTS

ZL is supported by Peking University under BICMR mathematical scholarship. HJ is supported by National University of Singapore under PGF scholarship. QL is supported by the National Research Foundation, Singapore, under the NRF fellowship (NRF-NRFF13-2021-0005).

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

## A    Properties of model functionals

In this section, we prove observations of the hypothesis space reported in Proposition 3.1.

*Proof of Proposition 3.1.* Recall that

$$\widehat{H}_t(\boldsymbol{x};\theta) = \int_0^\infty c^\top e^{Vt} M e^{Ws} U x_{-s} ds. \tag{12}$$

Fix any $\theta = (W, V, U, M, c)$. The linearity is obvious. Since both $W \in \mathcal{W}_{m_E}$ and $V \in \mathcal{V}_{m_D}$ have eigenvalues with negative real parts, there exist $c_1, c_2, c_1', c_2' > 0$, such that $\|e^{Vt}\|_\infty \leq c_1 e^{-c_2 t}$ and $\|e^{Ws}\|_\infty \leq c_1' e^{-c_2' s}$ for any $t, s \geq 0$, hence

$$
\begin{aligned}
|c^\top e^{Vt} M e^{Ws} U x_{-s}| &\leq \|c\|_1 \|e^{Vt} M e^{Ws} U x_{-s}\|_\infty \\
&\leq \|c\|_1 \|e^{Vt}\|_\infty \|M\|_\infty \|e^{Ws}\|_\infty \|U\|_\infty \|x_{-s}\|_\infty \\
&\lesssim e^{-c_2 t} e^{-c_2' s} \|\boldsymbol{x}\|_\mathcal{X},
\end{aligned}
\tag{13}
$$

where $\lesssim$ hides universal positive constants depending on parameters $\theta$. Therefore

$$|\widehat{H}_t(\boldsymbol{x};\theta)| \lesssim e^{-c_2 t} \|\boldsymbol{x}\|_\mathcal{X} \Rightarrow \|\widehat{H}_t(\cdot;\theta)\| \lesssim e^{-c_2 t}. \tag{14}$$

That is, the functional $\widehat{H}_t(\cdot;\theta)$ is bounded (i.e. continuous) with an exponentially-decayed norm (as a function of $t$). Finally, by (13) and Lebesgue's dominated convergence theorem, the functional $\widehat{H}_t(\cdot;\theta)$ is also regular. The proof is completed. $\square$

## B    Universal approximation

In this section, we provide the proof of Theorem 4.1, i.e. the universal approximation property of RNN encoder-decoders. As is stated in the main text, the key step is to utilise the classical representation theorem, which helps us to reduce the approximation problem of functionals to functions.

### B.1    Preliminaries

First, we list the background definitions and notations used in the following theorems. Let $(X, \mathcal{A})$ be a measure space (with $\mathcal{A}$ as the $\sigma$-algebra of subsets of $X$).

- The space $X$ is called *locally compact* if for any $x \in X$, $x$ has a compact neighbourhood.
- $X$ is a *Hausdorff space* if all distinct points in $X$ are pair-wisely separable by neighbourhoods. That is, for any $x, y \in X$, there exists a neighbourhood $\Delta_x$ of $x$ and a neighbourhood $\Delta_y$ of $y$, such that $\Delta_x \cap \Delta_y = \varnothing$.
- The measure $\mu$ is called a *finite measure*, if it satisfies $\mu(X) < \infty$. The measure $\nu$ is called a *$\sigma$-finite measure*, if $X$ can be covered with at most countably many measurable sets with finite measure. That is, there are measurable sets $\{A_n\}_{n=1}^\infty \subset \mathcal{A}$ with $\nu(A_n) < \infty$ for all $n \in \mathbb{N}_+$, such that $\bigcup_{n=1}^\infty A_n = X$. Obviously, a finite measure is also $\sigma$-finite.
- Let $X = (-\infty, 0]$. For a measure $\mu$ on the measure space $((-\infty, 0], \mathcal{A})$, $\mu$ is *absolutely continuous* with respect to the Lebesgue measure $\nu$, if for every measurable set $A$, $\nu(A) = 0$ implies $\mu(A) = 0$, which is written as $\mu \ll \nu$.

Denote by $C_0(X)$ the linear space of continuous functions defined on $X$ vanishing at infinity. We have the following classical representation theorem.

**Theorem B.1** (Riesz-Markov-Kakutani representation theorem)**.** *Let $X$ be a locally compact Hausdorff space. For any continuous linear functional $\psi$ on $C_0(X)$, there is a unique, regular, countably additive and signed measure $\mu$ on $X$, such that*

$$\psi(f) = \int_X f(x) d\mu(x), \quad \forall f \in C_0(X), \tag{15}$$

with $\|\psi\| = |\mu|(X)$. Here, $|\mu|(X)$ denotes the total variation of (the signed measure) $\mu$, which is defined as $|\mu|(X) := \sup_{\mathcal{P}} \sum_{i=1}^{k} |\mu(A_i)|$, where $\mathcal{P}: X = \bigcup_{i=1}^{k} A_i$ is a partition over $X$ with $A_i \in \mathcal{A}$ for all $i = 1, 2, \cdots, k$.

*Proof.* Well-known, see e.g. Bogachev (2007) (CH 7.10.4). $\qquad\square$

**Remark B.1.** *It is straightforward to verify that $|\mu|(X) = \sup_{A \in \mathcal{A}}(|\mu(A)| + |\mu(A^c)|)$. Furthermore, if $\mu$ has a density $d\mu/d\nu$ with respect to a countably additive, nonnegative measure $\nu$, then we have $|\mu|(X) = \|d\mu/d\nu\|_{L^1(\nu)}$.*

To handle signed measures, the following Jordan decomposition theorem (Bogachev, 2007) is necessary.

**Theorem B.2.** *Let $\mu$ be a signed measure on the measure space $(X, \mathcal{A})$. Then, there are two mutually singular (non-negative) measures $\mu^+$ and $\mu^-$ on $(X, \mathcal{A})$, such that $\mu = \mu^+ - \mu^-$. Moreover, such a pair $(\mu^+, \mu^-)$ is unique.*

Based on this, we have the following proposition to characterise absolutely continuous signed measures.

**Proposition B.1.** *If $\mu$ and $\nu$ are signed measures, then we have $\mu \ll \nu \Leftrightarrow \mu^+ \ll \nu$ and $\mu^- \ll \nu$.*

We also need the following Radon-Nikodym theorem (Bogachev, 2007).

**Theorem B.3.** *Let $(X, \mathcal{A}, \nu)$ be a $\sigma$-finite measure space, and let $\mu$ be a $\sigma$-finite signed measure, such that $\mu \ll \nu$. Then there exists a unique measurable function $f$, such that $\mu(A) = \int_A f d\nu$ for every measurable set $A$.*

## B.2 Proofs

Before we prove the universal approximation theorem (Theorem 4.1), we need some lemmas.

**Lemma B.1.** *Let $\{H_t : t \geq 0\}$ be a family of linear, continuous and regular functionals defined on $\mathcal{X}$. Then there exists a integrable function $\rho : [0, \infty)^2 \to \mathbb{R}^d$, i.e.*

$$\|\rho\|_{L^1([0,\infty)^2)} := \sum_{i=1}^{d} \|\rho_i\|_{L^1([0,\infty)^2)} < \infty, \qquad (16)$$

*such that*

$$H_t(\boldsymbol{x}) = \int_0^\infty x_{-s}^\top \rho(t, s) ds, \quad \forall \boldsymbol{x} \in \mathcal{X}. \qquad (17)$$

*In particular, we have $\|\boldsymbol{H}\| = \int_0^\infty \|H_t\| dt = \|\rho\|_{L^1([0,\infty)^2)}$.*

*Proof.* Obviously, $(-\infty, 0]$ is a locally compact Hausdorff space. For any $t \geq 0$, since $H_t$ is linear continuous, according to the Riesz-Markov-Kakutani representation theorem (Theorem B.1), there exists a unique, regular, countably additive and signed measure $\mu_t$, such that

$$H_t(\boldsymbol{x}) = \int_{-\infty}^{0} x_s^\top d\mu_t(s), \quad \forall \boldsymbol{x} \in \mathcal{X}, \qquad (18)$$

with $\sum_{i=1}^{d} |\mu_{t,i}|((-\infty, 0]) = \|H_t\|$. We show that for any $t \geq 0$, $i = 1, 2, \cdots, d$, $\mu_{t,i}$ is absolutely continuous with respect to $\nu$ (the Lebesgue measure), i.e. $\mu_{t,i} \ll \nu$. According to Theorem B.2 and Proposition B.1, one can assume $\mu_{t,i}$ to be non-negative without loss of generality. Take a measurable set $A \subset (-\infty, 0]$ with $\nu(A) = 0$, the aim is to show $\mu_{t,i}(A) = 0$. Let $A' = (-\infty, 0] \setminus A$. Since both $A$ and $A'$ are measurable, there exist $K_n \subset A$, $K'_n \subset A'$ with $K_n, K'_n$ closed, such that $\mu_{t,i}(A \setminus K_n) \leq 1/n$, $\mu_{t,i}(A' \setminus K'_n) \leq 1/n$

and $\nu(A' \setminus K'_n) \leq 1/n$ for any $n \in \mathbb{N}_+$. Fix any $i \in \{1, 2, \ldots, d\}$, we construct the sequence of input signals $\{\boldsymbol{x}^{(n)}\}_{n=1}^{\infty}$ as

$$x_{s,j}^{(n)} = \begin{cases} 0, & s \leq 0, \ j \neq i, \\ 0, & s \in K'_n, \ j = i, \\ 1, & s \in K_n, \ j = i, \end{cases} \qquad j = 1, 2, \ldots, d, \tag{19}$$

which can then be continuously extended to $(-\infty, 0]$ by defining $x_{s,i}^{(n)} := \frac{d(s, K'_n)}{d(s, K_n) + d(s, K'_n)} \in [0, 1]$. [1]

We deduce that $\lim_{n \to \infty} x_{s,i}^{(n)} = 0$ for $\nu$-a.e. $s \leq 0$. In fact, let $S := \{s \leq 0 : \lim_{n \to \infty} x_{s,i}^{(n)} = 0\}$, we have $K'_n \subset S$ since for any $s \in K'_n$, $x_{s,i}^{(n)} = 0$. Hence, $(-\infty, 0] \setminus S \subset A \cup (A' \setminus K'_n)$, which gives $\nu((-\infty, 0] \setminus S) \leq \nu(A) + \nu(A' \setminus K'_n) \leq 1/n \to 0$ as $n \to \infty$. Due to the regularity of $H_t$, we get $\lim_{n \to \infty} H_t(\boldsymbol{x}^{(n)}) = 0$. By (18) and (19), we have

$$H_t(\boldsymbol{x}^{(n)}) = \sum_{j=1}^{d} \int_{-\infty}^{0} x_{s,j}^{(n)} d\mu_{t,j}(s) = \int_{-\infty}^{0} x_{s,i}^{(n)} d\mu_{t,i}(s)$$

$$= \int_{K_n} + \int_{A \setminus K_n} + \int_{A' \setminus K'_n} x_{s,i}^{(n)} d\mu_{t,i}(s) = \mu_{t,i}(K_n) + I_{1,n} + I_{2,n}, \tag{20}$$

where $\mu_{t,i}(K_n) = \mu_{t,i}(A) - \mu_{t,i}(A \setminus K_n) \in [\mu_{t,i}(A) - 1/n, \mu_{t,i}(A)]$, and $|I_{1,n}| + |I_{2,n}| \leq \int_{A \setminus K_n} + \int_{A' \setminus K'_n} 1 d\mu_{t,i}(s) = \mu_{t,i}(A \setminus K_n) + \mu_{t,i}(A' \setminus K'_n) \leq 2/n$, which gives $\lim_{n \to \infty} H_t(\boldsymbol{x}^{(n)}) = \mu_{t,i}(A)$. Therefore, $\mu_{t,i}(A) = 0$.

Notice that $|\mu_{t,i}((-\infty, 0])| \leq |\mu_{t,i}|((-\infty, 0]) \leq \|H_t\| < \infty$ for a.e. $t \geq 0$ (since $\|\boldsymbol{H}\| = \int_0^{\infty} \|H_t\| dt < \infty$), we get that $((-\infty, 0], \mathcal{A}, \mu_{t,i})$ is a finite measure space, and hence $\sigma$-finite. Obviously, $((-\infty, 0], \mathcal{A}, \nu)$ is a $\sigma$-finite measure space. According to the Radon-Nikodym theorem (Theorem B.3), there exists a unique measurable function $\rho_{t,i} : (-\infty, 0] \to \mathbb{R}$, such that $\mu_{t,i}(A) = \int_A \rho_{t,i}(s) d\nu(s)$ for every measurable set $A$. Hence, we have

$$H_t(\boldsymbol{x}) = \int_{-\infty}^{0} x_s^{\top} \rho_t(s) ds = \int_{0}^{\infty} x_{-s}^{\top} \rho(t, s) ds, \quad \forall \boldsymbol{x} \in \mathcal{X}, \tag{21}$$

with $\rho(t, s) := \rho_t(-s)$. In addition, by Remark B.1, we have $|\mu_{t,i}|((-\infty, 0]) = \int_{-\infty}^{0} |\rho_{t,i}(s)| ds$, which gives

$$\|\boldsymbol{H}\| = \int_0^{\infty} \|H_t\| dt = \sum_{i=1}^{d} \int_0^{\infty} |\mu_{t,i}|((-\infty, 0]) dt = \sum_{i=1}^{d} \|\rho_i\|_{L^1([0,\infty)^2)} = \|\rho\|_{L^1([0,\infty)^2)}. \tag{22}$$

The proof is completed. $\qquad\qquad\qquad\qquad\qquad\qquad\qquad\qquad\qquad\qquad\qquad\qquad\qquad\square$

Based on this representation theorem, the problem of functional approximation is reduced as function approximation. That is,

$$\|\boldsymbol{H} - \widehat{\boldsymbol{H}}\| = \int_0^{\infty} \|H_t - \widehat{H}_t\| dt = \int_0^{\infty} \sup_{\|\boldsymbol{x}\|_{\mathcal{X}} \leq 1} \left| H_t(\boldsymbol{x}) - \widehat{H}_t(\boldsymbol{x}; \theta) \right| dt$$

$$= \int_0^{\infty} \sup_{\|\boldsymbol{x}\|_{\mathcal{X}} \leq 1} \left| \int_0^{\infty} x_{-s}^{\top} (\rho(t, s) - \hat{\rho}(t, s)) ds \right| dt$$

$$\leq \int_0^{\infty} \sup_{\|\boldsymbol{x}\|_{\mathcal{X}} \leq 1} \int_0^{\infty} \|x_{-s}\|_{\infty} \|\rho(t, s) - \hat{\rho}(t, s)\|_1 ds dt$$

$$\leq \sum_{i=1}^{d} \int_0^{\infty} \int_0^{\infty} |\rho_i(t, s) - \hat{\rho}_i(t, s)| ds dt, \tag{23}$$

_______________
[1] Here, $d(s, B) := \inf\{|s - a| : a \in B\}$ is the distance between a point $s$ and a set $B$.

i.e.

$$\|\boldsymbol{H} - \widehat{\boldsymbol{H}}\| \leq \|\rho - \hat{\rho}\|_{L^1([0,\infty)^2)} := \sum_{i=1}^{d} \|\rho_i - \hat{\rho}_i\|_{L^1([0,\infty)^2)}. \tag{24}$$

**Lemma B.2.** *Let $\rho(t,s) : [0,\infty)^2 \to \mathbb{R}$ with $\|\rho\|_{L^1([0,\infty)^2)} < \infty$. Then for any $\epsilon > 0$, there exists a polynomial $p(u,v) = \sum_{j=1,k=1}^{m} c_{jk} u^j v^k$, such that*

$$\int_0^\infty \int_0^\infty |\rho(t,s) - p(e^{-t}, e^{-s})| dt ds < \epsilon. \tag{25}$$

*Proof.* Fix any $\epsilon > 0$. Consider the following transformation

$$R(u,v) = \begin{cases} \frac{1}{uv} \rho(-\ln u, -\ln v), & u, v \in (0,1], \\ 0, & uv = 0. \end{cases} \tag{26}$$

This transformation preserves the norm with $\|\rho\|_{L^1([0,\infty)^2)} = \|R\|_{L^1([0,1]^2)}$.

First, according to the density of continuous functions in $L^p$ space (Rudin, 1987, Theorem 3.14), there exists $\tilde{R} \in C([0,1]^2)$, such that $\|R - \tilde{R}\|_{L^1([0,1]^2)} < \epsilon/2$. Next, by the density of polynomials in the space of continuous functions (Lorentz, 2005, Theorem 6), there exists a polynomial $q(u,v) = \sum_{j,k=0}^{m} c_{jk} u^j v^k$, such that $\|\tilde{R} - q\|_{L^\infty([0,1]^2)} < \epsilon/2$. Finally, let $p(u,v) = uvq(u,v)$, we have

$$\begin{aligned}
\int_0^\infty \int_0^\infty |\rho(t,s) - p(e^{-t}, e^{-s})| dt ds &= \int_0^1 \int_0^1 \left| R(u,v) - \frac{1}{uv} p(u,v) \right| du dv \\
&= \|R - q\|_{L^1([0,1]^2)} \\
&\leq \|R - \tilde{R}\|_{L^1([0,1]^2)} + \|\tilde{R} - q\|_{L^\infty([0,1]^2)} \\
&< \epsilon/2 + \epsilon/2 = \epsilon,
\end{aligned} \tag{27}$$

which completes the proof. $\square$

Now we are ready to prove the universal approximation theorem.

*Proof of Theorem 4.1.* According to the representation theorem (Lemma B.1), we have

$$H_t(\boldsymbol{x}) = \int_0^\infty x_{-s}^\top \rho(t,s) ds, \quad \forall \boldsymbol{x} \in \mathcal{X}, \tag{28}$$

where $\|\rho\|_{L^1([0,\infty)^2)} = \|\boldsymbol{H}\| < \infty$. Therefore, by Lemma B.2, there exists $p_i(u,v) = \sum_{j,k=1}^{m} c_{jk}^{(i)} u^j v^k$, $i = 1, 2, \ldots, d$, where $m$ is the maximal degree of $\{p_i\}_{i=1}^d$, such that

$$\sum_{i=1}^{d} \int_0^\infty \int_0^\infty |\rho_i(t,s) - p_i(e^{-t}, e^{-s})| dt ds < \epsilon. \tag{29}$$

Let

$$\begin{aligned}
c &= u = \mathbf{1}_m, \, V = \tilde{W} = -\operatorname{diag}(1, 2, \ldots, m), \\
W &= \operatorname{diag}(\tilde{W}, \tilde{W}, \cdots, \tilde{W}) \in \mathbb{R}^{dm \times dm}, \, U = \operatorname{diag}(u, u, \cdots, u) \in \mathbb{R}^{dm \times d}, \\
M &= PQ = (M_1, M_2, \cdots, M_d) \in \mathbb{R}^{m \times dm}, \, [M_i]_{jk} = c_{jk}^{(i)},
\end{aligned} \tag{30}$$

we get

$$\begin{aligned}
\hat{\rho}(t,s)^\top &= c^\top e^{Vt} PQ e^{Ws} U = c^\top e^{Vt} M e^{Ws} U \\
&= c^\top e^{Vt} \cdot (M_1, M_2, \cdots, M_d) \cdot \operatorname{diag}(e^{Ws}, e^{Ws}, \cdots, e^{Ws}) \cdot \operatorname{diag}(u, u, \cdots, u) \\
&= (c^\top e^{Vt} M_1, c^\top e^{Vt} M_2, \cdots, c^\top e^{Vt} M_d) \cdot \operatorname{diag}(e^{Ws} u, e^{Ws} u, \cdots, e^{Ws} u) \\
&= (c^\top e^{Vt} M_1 e^{Ws} u, c^\top e^{Vt} M_2 e^{Ws} u, \cdots, c^\top e^{Vt} M_d e^{Ws} u),
\end{aligned} \tag{31}$$

with

$$\hat{\rho}_i(t,s) = c^\top e^{Vt} M_i e^{Ws} u = p_i(e^{-t}, e^{-s}), \quad i = 1, 2, \ldots, d. \tag{32}$$

Therefore, by (24) and (29), we have

$$\|\boldsymbol{H} - \widehat{\boldsymbol{H}}\| \leq \sum_{i=1}^d \|\rho_i - \hat{\rho}_i\|_{L^1([0,\infty)^2)}$$

$$= \sum_{i=1}^d \int_0^\infty \int_0^\infty \left|\rho_i(t,s) - p_i(e^{-t}, e^{-s})\right| ds dt < \epsilon, \tag{33}$$

which completes the proof. $\qquad\square$

## C  GENERAL APPROXIMATION RATES

In this section, the proof of Theorem 4.2 is given. Again, by (24), the aim now is to investigate the function approximation $\|\rho - \hat{\rho}\|$. Since one can handle each spatial dimension separately (similarly with (30) and (31)), we firstly derive the estimates by assuming $d = 1$, and then extend the obtained results to the case of multi-dimensional inputs (for general $d \in \mathbb{N}_+$).

**Conditions on representation.** To characterise the accuracy of using the model $c^\top e^{Vt} M e^{Ws} u$ (with $M := PQ$) to approximate the target $\rho(t,s)$, the first stuff is to translate the conditions on the output (of piece-wise constant signals) to the representation. Recall that $y^c(t,s) = H_t(\mathbf{1}_{(-\infty,-s]})$, $t, s \geq 0$, we get $\rho(t,s) = -\frac{d}{ds} H_t(\mathbf{1}_{(-\infty,-s]})$. Hence, the assumptions on $y^c$ in Theorem 4.2 is equivalent to the following smoothness and exponential decay conditions on $\rho$. That is, there exist $\alpha \in \mathbb{N}_+$, $\beta > 0$ such that

$$\rho \in C^\alpha([0, +\infty)^2), \tag{34}$$

$$e^{\beta(t+s)} \frac{\partial^{k+l}}{\partial t^k \partial s^l} \rho(t,s) = o(1) \text{ as } \|(t,s)\| \to \infty, \quad k, l \in \mathbb{N}, \ k+l \leq \alpha. \tag{35}$$

Note that the last decay condition implies

$$\sup_{t,s \geq 0} \beta^{-(k+l)} e^{\beta(t+s)} \left|\frac{\partial^{k+l}}{\partial t^k \partial s^l} \rho(t,s)\right| \leq \gamma, \quad k, l \in \mathbb{N}, \ k+l \leq \alpha \tag{36}$$

for some $\gamma > 0$.

### C.1  BASICS

Let $\Omega \in \mathbb{R}^d$ be a bounded set. Define the spaces

$$C^\alpha(\Omega) := \{f \in C(\bar{\Omega}) : D^{\boldsymbol{i}} f \in C(\bar{\Omega}) \text{ for all } |\boldsymbol{i}| \leq \alpha\}, \quad \alpha \in \mathbb{N}, \tag{37}$$

$$C^{\alpha,\mu}(\Omega) := \{f \in C^\alpha(\Omega) : |D^{\boldsymbol{i}} f(x) - D^{\boldsymbol{i}} f(y)| \leq K\|x - y\|_2^\mu \text{ for some } K > 0,$$
$$\text{for all } x, y \in \Omega \text{ and } |\boldsymbol{i}| = \alpha\}, \tag{38}$$

and the "norm"

$$|f|_{\alpha,\mu,\Omega} := \sup_{|\boldsymbol{i}|=\alpha} \sup_{x,y \in \Omega} \frac{|D^{\boldsymbol{i}} f(x) - D^{\boldsymbol{i}} f(y)|}{\|x - y\|_2^\mu}, \quad \forall f \in C^{\alpha,\mu}(\Omega), \tag{39}$$

with the shorthand $\|\cdot\|_\Omega := |\cdot|_{0,0,\Omega}$.

**Theorem C.1** (Multivariate Jackson's theorem (Schultz, 1969), Theorem 4.10). *Let $\Omega \in \mathbb{R}^d$ be a regular,* [2] *bounded and open set, and $f \in C^{\alpha,\mu}(\bar{\Omega})$ for some $\alpha \in \mathbb{N}$, $\mu \in [0,1]$. Then for any $n \in \mathbb{N}_+$, we have*

$$\inf_{p \in \mathcal{P}_n^d} \|f - p\|_{\bar{\Omega}} \leq \frac{C(\alpha, \mu)}{n^{\alpha+\mu}} |f|_{\alpha,\mu,\bar{\Omega}}, \tag{40}$$

---

[2]It is proved that every bounded, open and convex set is regular. See Morrey (1966) (Lemma 3.4.1).

where $\mathcal{P}_n^d$ denotes the set of all polynomials with the degree of no more than $n$ in each variable, $C(\alpha, \mu) > 0$ is a universal constant only depending on $\alpha, \mu$ and $\Omega$.

A commonly used case is when $\mu = 0$. That is, for $f \in C^\alpha(\bar{\Omega})$, we get

$$|f|_{\alpha,0,\bar{\Omega}} \le 2 \max_{|\boldsymbol{i}|=\alpha} \|D^{\boldsymbol{i}} f\|_{L^\infty(\bar{\Omega})} < \infty. \tag{41}$$

For any $x, x_0 \in \bar{\Omega}$, let $\tilde{p}(x) := p(x) + f(x_0) - p(x_0)$, and $\tilde{e}(x) := f(x) - \tilde{p}(x)$. Then $\tilde{p} \in \mathcal{P}_n^d$, and

$$\begin{aligned}
|f(x) - \tilde{p}(x)| &\le |\tilde{e}(x_0)| + |\tilde{e}(x) - \tilde{e}(x_0)| \\
&\le \sup_{x,y \in \bar{\Omega}} |\tilde{e}(x) - \tilde{e}(y)| = |\tilde{e}|_{0,0,\bar{\Omega}} = \|f - \tilde{p}\|_{\bar{\Omega}}, \quad \forall x \in \bar{\Omega}.
\end{aligned} \tag{42}$$

This gives the following convenient corollary.

**Corollary C.1.** *Let $\Omega \in \mathbb{R}^d$ be a regular, bounded and open set, and $f \in C^\alpha(\bar{\Omega})$ for some $\alpha \in \mathbb{N}$. Then for any $n \in \mathbb{N}_+$, there exists $p \in \mathcal{P}_n^d$ such that*

$$\|f - p\|_{L^\infty(\bar{\Omega})} \le \frac{C_\alpha}{n^\alpha} \max_{|\boldsymbol{i}|=\alpha} \|D^{\boldsymbol{i}} f\|_{L^\infty(\bar{\Omega})}, \tag{43}$$

*where $\mathcal{P}_n^d$ denotes the set of all polynomials with the degree of no more than $n$ in each variable, $C_\alpha > 0$ is a universal constant only depending on $\alpha$ and $\Omega$.*

### C.2 Proofs

Now we are ready to present the proof.

*Proof of Theorem 4.2. Step 1: domain transform.* Consider the transform from the infinite domain $[0, \infty)^2$ to the compact one $[0, 1]^2$:

$$R(u, v) = \begin{cases} \frac{1}{uv} \rho(-c_0 \ln u, -c_0 \ln v), & u, v \in (0, 1], \\ 0, & uv = 0, \end{cases} \tag{44}$$

where $c_0 := (\alpha + 1)/\beta > 0$ is a fixed constant. A straightforward computation by induction shows that, for any $k, l \in \mathbb{N}$, $k + l \le \alpha$, and any $u, v \in (0, 1]$,

$$\frac{\partial^{k+l}}{\partial u^k \partial v^l} R(u, v) = \frac{(-1)^{k+l}}{u^{k+1} v^{l+1}} \sum_{i=0}^{k} \sum_{j=0}^{l} C(k,i) C'(l,j) c_0^{i+j} \frac{\partial^{i+j}}{\partial t^i \partial s^j} \rho(-c_0 \ln u, -c_0 \ln v), \tag{45}$$

where $C(k, i), C'(l, j)$ are some integer constants, and $(t, s) = (-c_0 \ln u, -c_0 \ln v)$ is a one-to-one mapping between $(0, 1]^2$ and $[0, \infty)^2$. By (36), we get

$$\left| \frac{\partial^{k+l}}{\partial u^k \partial v^l} R(u, v) \right| \le \frac{1}{u^{k+1} v^{l+1}} \sum_{i=0}^{k} \sum_{j=0}^{l} |C(k,i)||C'(l,j)| c_0^{i+j} \left| \frac{\partial^{i+j}}{\partial t^i \partial s^j} \rho(-c_0 \ln u, -c_0 \ln v) \right|,$$

$$\left| \frac{\partial^{k+l}}{\partial u^k \partial v^l} R(e^{-\frac{t}{c_0}}, e^{-\frac{s}{c_0}}) \right| \le e^{\frac{(k+1)}{c_0} t} e^{\frac{(l+1)}{c_0} s} \sum_{i=0}^{k} \sum_{j=0}^{l} |C(k,i)||C'(l,j)| c_0^{i+j} \left| \frac{\partial^{i+j}}{\partial t^i \partial s^j} \rho(t, s) \right|$$

$$\le \sum_{i=0}^{k} \sum_{j=0}^{l} |C(k,i)||C'(l,j)|(\alpha+1)^{i+j} \cdot \beta^{-(i+j)} e^{\beta(t+s)} \left| \frac{\partial^{i+j}}{\partial t^i \partial s^j} \rho(t, s) \right|$$

$$\le \sum_{i=0}^{k} \sum_{j=0}^{l} |C(k,i)||C'(l,j)|(\alpha+1)^{i+j} \gamma \le C(\alpha)\gamma, \tag{46}$$

where $C(\alpha) > 0$ is a universal constant only depending on $\alpha$. According to the decay condition (35), we get

$$\lim_{(u,v)\to(0,0)} \frac{\frac{\partial^{i+j}}{\partial t^i \partial s^j}\rho(-c_0 \ln u, -c_0 \ln v)}{u^{k+1}v^{l+1}} = \lim_{(t,s)\to(+\infty,+\infty)} e^{\frac{(k+1)}{c_0}t}e^{\frac{(l+1)}{c_0}s}\frac{\partial^{i+j}}{\partial t^i \partial s^j}\rho(t,s)$$

$$= \lim_{(t,s)\to(+\infty,+\infty)} e^{\frac{(k-\alpha)}{\alpha+1}\beta t}e^{\frac{(l-\alpha)}{\alpha+1}\beta s}\cdot e^{\beta(t+s)}\frac{\partial^{i+j}}{\partial t^i \partial s^j}\rho(t,s)$$

$$= 0, \tag{47}$$

and similarly for $u_0, v_0 \in (0,1]$,

$$\lim_{(u,v)\to(u_0,0)} \frac{\frac{\partial^{i+j}}{\partial t^i \partial s^j}\rho(-c_0 \ln u, -c_0 \ln v)}{u^{k+1}v^{l+1}} = \lim_{(t,s)\to(-c_0 \ln u_0,+\infty)} e^{\frac{(k+1)}{c_0}t}e^{\frac{(l+1)}{c_0}s}\frac{\partial^{i+j}}{\partial t^i \partial s^j}\rho(t,s)$$

$$= \lim_{\|(t,s)\|\to+\infty} e^{\frac{(k-\alpha)}{\alpha+1}\beta t}e^{\frac{(l-\alpha)}{\alpha+1}\beta s}\cdot e^{\beta(t+s)}\frac{\partial^{i+j}}{\partial t^i \partial s^j}\rho(t,s)$$

$$= 0, \tag{48}$$

$$\lim_{(u,v)\to(0,v_0)} \frac{\frac{\partial^{i+j}}{\partial t^i \partial s^j}\rho(-c_0 \ln u, -c_0 \ln v)}{u^{k+1}v^{l+1}} = 0. \tag{49}$$

This gives

$$\frac{\partial^{k+l}}{\partial u^k \partial v^l}R(u,v) = 0, \quad (u,v)\in[0,1]\times\{0\}\cup\{0\}\times[0,1], \tag{50}$$

and

$$M_0 := \max_{k,l\in\mathbb{N},\,k+l\le\alpha}\max_{(u,v)\in[0,1]^2}\left|\frac{\partial^{k+l}}{\partial u^k \partial v^l}R(u,v)\right| \le C(\alpha)\gamma \tag{51}$$

by (46) and (50). Hence, $R(u,v)\in C^\alpha([0,1]^2)$ with bounded derivatives.

*Step 2: polynomial approximation.* According to Corollary C.1 and (51), we obtain that there exists $\tilde{R}_n \in \mathcal{P}_n^2$, such that

$$\|R - \tilde{R}_n\|_{L^\infty([0,1]^2)} \le \frac{C_\alpha}{n^\alpha}\max_{k,l\in\mathbb{N},\,k+l=\alpha}\left\|\frac{\partial^{k+l}}{\partial u^k \partial v^l}R(u,v)\right\|_{L^\infty([0,1]^2)}$$

$$\le \frac{C(\alpha)\gamma}{n^\alpha}, \quad \forall n\in\mathbb{N}_+, \tag{52}$$

where $C(\alpha) > 0$ is a universal constant only related to $\alpha$. Furthermore, let $\hat{R}_n(u,v) := \tilde{R}_n(u,v) - \tilde{R}_n(u,0) - \tilde{R}_n(0,v) + \tilde{R}_n(0,0)$, we get $\hat{R}_n \in \mathcal{P}_n^2$ with $\hat{R}_n(u,0) = \hat{R}_n(0,v) = 0$ for any $u,v\in[0,1]$. By (50), we have $R(u,v) = 0$ for any $(u,v)\in[0,1]\times\{0\}\cup\{0\}\times[0,1]$, then

$$\|R - \hat{R}_n\|_{L^\infty([0,1]^2)} \le \|R - \tilde{R}_n\|_{L^\infty([0,1]^2)} + \|\tilde{R}_n - \hat{R}_n\|_{L^\infty([0,1]^2)}$$

$$\le \|R - \tilde{R}_n\|_{L^\infty([0,1]^2)} + \|\tilde{R}_n(u,0) - R(u,0)\|_{L^\infty([0,1]^2)}$$

$$+ \|\tilde{R}_n(0,v) - R(0,v)\|_{L^\infty([0,1]^2)} + \|\tilde{R}_n(0,0) - R(0,0)\|_{L^\infty([0,1]^2)}$$

$$\lesssim \|R - \tilde{R}_n\|_{L^\infty([0,1]^2)}, \tag{53}$$

i.e. we can further require the approximator satisfying the zero half-boundary condition (i.e. vanishing on $(u,v)\in[0,1]\times\{0\}\cup\{0\}\times[0,1]$) without effecting the approximation accuracy. Let $\bar{m} := \min\{m_E, m_D\}$, we get

$$\|R - \tilde{R}\|_{L^\infty([0,1]^2)} \le \frac{C(\alpha)\gamma}{\bar{m}^\alpha} \le C(\alpha)\gamma\left(\frac{1}{m_E^\alpha} + \frac{1}{m_D^\alpha}\right), \tag{54}$$

where

$$\tilde{R} := \tilde{R}_{\bar{m}} \in \mathcal{P}_{\bar{m}}^2, \quad \tilde{R}(u,v) := \sum_{i=1}^{\bar{m}}\sum_{j=1}^{\bar{m}}\tilde{r}_{ij}u^i v^j. \tag{55}$$

Let

$$c = \mathbf{1}_{\bar{m}}, \quad u = \mathbf{1}_{\bar{m}}, \quad M = [\tilde{r}_{ij}] \in \mathbb{R}^{\bar{m} \times \bar{m}}, \tag{56}$$

$$V = -\mathrm{diag}(2, 3, \cdots, \bar{m} + 1)/c_0, \quad W = -\mathrm{diag}(2, 3, \cdots, \bar{m} + 1)/c_0, \tag{57}$$

then by (24) and (54), we have

$$\|\boldsymbol{H} - \widehat{\boldsymbol{H}}\| \leq \|\rho - \hat{\rho}\|_{L^1([0,\infty)^2)} \tag{58}$$

$$= \left\|\rho(t,s) - c^\top e^{Vt} M e^{Ws} u\right\|_{L^1([0,\infty)^2)}$$

$$= \left\|\rho(t,s) - e^{-\frac{t}{c_0}} e^{-\frac{s}{c_0}} \tilde{R}(e^{-\frac{t}{c_0}}, e^{-\frac{s}{c_0}})\right\|_{L^1([0,\infty)^2)}$$

$$= c_0^2 \left\|R - \tilde{R}\right\|_{L^1([0,1]^2)} \tag{59}$$

$$\leq c_0^2 \left\|R - \tilde{R}\right\|_{L^\infty([0,1]^2)} \leq \frac{C(\alpha)\gamma}{\beta^2 \bar{m}^\alpha} \leq \frac{C(\alpha)\gamma}{\beta^2} \left(\frac{1}{m_E^\alpha} + \frac{1}{m_D^\alpha}\right). \tag{60}$$

That is, one can achieve an approximation accuracy scaling like $(1/\bar{m})^\alpha$ with $\bar{m}^2$ parameters. The proof is completed. $\qquad \square$

**Remark C.1.** *The extension to multi-dimensional inputs (general $d \in \mathbb{N}_+$) is found in the last paragraph of Appendix D.2.*

**Remark C.2.** *Recall $M = PQ \in \mathbb{R}^{m_D \times m_E}$ with $P \in \mathbb{R}^{m_D \times N}$, $Q \in \mathbb{R}^{N \times m_E}$, we only need to investigate the case of $N \leq \min\{m_E, m_D\} = \bar{m}$, since $\mathrm{rank}(M) \leq \bar{m}$.*

## D Approximation rates via temporal product structure

In this section, the proof of Theorem 4.3 is provided.

### D.1 Proper Orthogonal Decomposition

Proper orthogonal decomposition (POD; (Liang et al., 2002), (Berkooz et al., 1993), (Chatterjee, 2000)) is a method for model reduction, which is commonly applied to numerical PDEs and fluids mechanics. It can be viewed as an extension of singular value decomposition (SVD) and principal component analysis (PCA) to infinite-dimensional spaces.

Fix any $R \in L^\infty([0,1]^2)$. [3] Define the POD operator

$$\mathcal{K}: \phi(v) \mapsto \int_0^1 \int_0^1 R(u,v)\phi(v)dv \cdot R(u,v)du, \quad \phi(v) \in L^2[0,1]. \tag{61}$$

**Proposition D.1.** *The operator $\mathcal{K}$ is linear, bounded, compact, self-adjoint and non-negative.*

*Proof.* (i) The linearity is obvious.

(ii) Let

$$\mathcal{R}: \phi(v) \mapsto \int_0^1 R(u,v)\phi(v)dv, \quad \phi(v) \in L^2[0,1], \tag{62}$$

then

$$(\mathcal{K}\phi)(v) = \int_0^1 R(u,v)(\mathcal{R}\phi)(u)du, \quad \phi(v) \in L^2[0,1]. \tag{63}$$

By the Cauchy-Schwartz inequality, we get

$$\|\mathcal{R}\phi\|_{L^2[0,1]} \leq \|R\|_{L^2([0,1]^2)}\|\phi\|_{L^2[0,1]}, \tag{64}$$

---

[3]Here, we use the same notation as (44), since the function defined there is also bounded.

which gives

$$\|\mathcal{K}\phi\|_{L^2[0,1]} \leq \|R\|_{L^2([0,1]^2)}\|\mathcal{R}\phi\|_{L^2[0,1]} \leq \|R\|^2_{L^2([0,1]^2)}\|\phi\|_{L^2[0,1]}. \tag{65}$$

Note that $R(u,v) \in L^\infty([0,1]^2) \subset L^2([0,1]^2)$, hence both $\mathcal{K}$ and $\mathcal{R}$ are bounded operator from $L^2[0,1]$ to itself.

(iii) It is well-known that the Hilbert–Schmidt integral operator

$$(\mathcal{C}\psi)(v) = \int_0^1 R(u,v)\psi(u)du, \quad \psi(u) \in L^2[0,1] \tag{66}$$

is a compact operator from $L^2[0,1]$ to itself. Therefore

$$\mathcal{K}\phi = \mathcal{C}\mathcal{R}\phi, \quad \phi \in L^2[0,1], \tag{67}$$

which gives $\mathcal{K} = \mathcal{C}\mathcal{R}$. Since $\mathcal{R}$ is bounded and $\mathcal{C}$ is compact, we get $\mathcal{K}$ is also compact.

(iv) By Fubini's theorem, it is straightforward to verify that

$$\begin{aligned}
\langle \mathcal{K}\phi, \psi \rangle_{L^2[0,1]} &= \int_0^1 \int_0^1 R(u,w)\left(\int_0^1 R(u,v)\phi(v)dv\right)du \cdot \psi(w)dw \\
&= \int_0^1 \int_0^1 \int_0^1 R(u,w)R(u,v)\phi(v)\psi(w)dvdudw \\
&= \int_0^1 \int_0^1 R(u,v)\left(\int_0^1 R(u,w)\psi(w)dw\right)du \cdot \phi(v)dv \\
&= \langle \phi, \mathcal{K}\psi \rangle_{L^2[0,1]}, \quad \phi, \psi \in L^2[0,1].
\end{aligned} \tag{68}$$

(v) By Fubini's theorem, it is straightforward to verify that

$$\begin{aligned}
\langle \mathcal{K}\phi, \phi \rangle_{L^2[0,1]} &= \int_0^1 \int_0^1 (\mathcal{R}\phi)(u)R(u,w)du \cdot \phi(w)dw \\
&= \int_0^1 \int_0^1 (\mathcal{R}\phi)(u)R(u,w)\phi(w)dudw \\
&= \int_0^1 (\mathcal{R}\phi)(u)\left(\int_0^1 R(u,w)\phi(w)dw\right)du \\
&= \int_0^1 (\mathcal{R}\phi)^2(u)du \geq 0, \quad \phi \in L^2[0,1].
\end{aligned} \tag{69}$$

The proof is completed. $\qquad\square$

Combining (i)—(iv) and applying Hilbert–Schmidt's expansion theorem, we obtain that $L^2[0,1]$ has an orthonormal basis $\{\phi_n\}_{n\in\mathcal{N}}\bigcup\{\psi_\xi\}_{\xi\in\Xi}$, such that

- $\mathcal{K}\phi_n = \lambda_n\phi_n$, $\lambda_n \neq 0$ for $n \in \mathcal{N}$, and $\mathcal{K}\psi_\xi = 0$ for $\xi \in \Xi$, where $\mathcal{N}$ is a finite or countable set. If $\mathcal{N}$ is not finite, we have $\lim_{n\to\infty}\lambda_n = 0$;
- For any $\psi \in L^2[0,1]$, we have

$$\psi = \sum_{n\in\mathcal{N}} \langle \psi, \phi_n \rangle_{L^2[0,1]}\phi_n + \sum_{\xi\in\Xi}\langle \psi, \psi_\xi \rangle_{L^2[0,1]}\psi_\xi, \tag{70}$$

  where the second summation has at most countable non-zero terms, and

$$\mathcal{K}\psi = \sum_{n\in\mathcal{N}} \lambda_n \langle \psi, \phi_n \rangle_{L^2[0,1]}\phi_n. \tag{71}$$

Here, all the series converge under the norm $\|\cdot\|_{L^2[0,1]}$. Without loss of generality, $\mathcal{N} = \{1, 2, \cdots, N_0\}$ for $N_0 \in \mathbb{N}_+$ or $N_0 = +\infty$ (i.e. $\mathcal{N} = \mathbb{N}_+$). By (69), we get

$$0 \leq \langle \mathcal{K}\phi_n, \phi_n \rangle_{L^2[0,1]} = \lambda_n \langle \phi_n, \phi_n \rangle^2_{L^2[0,1]} = \lambda_n, \quad \forall n \in \mathcal{N}, \tag{72}$$

i.e. all the eigenvalues of $\mathcal{K}$ are non-negative, and $\lambda_n \neq 0$ for $n \in \mathcal{N}$ implies $\lambda_n > 0$, $\forall n \in \mathcal{N}$. In addition, $\lim_{n\to\infty} \lambda_n = 0$ implies that one can index all the eigenvalues in a non-increasing sequence: $\lambda_1 \geq \lambda_2 \geq \cdots \geq \lambda_n \geq \cdots \geq 0$.

Then, we can present the POD estimate.

**Theorem D.1.** *For any $R \in L^\infty([0,1]^2)$, we have*

$$\int_0^1 \left\| R(u,v) - \sum_{n=1}^N \langle R(u,v), \phi_n(v)\rangle_{L^2[0,1]} \phi_n(v) \right\|_{L^2[0,1]}^2 du = \sum_{n=N+1}^{N_0} \lambda_n, \quad \forall N \in \mathbb{N}. \tag{73}$$

*Proof.* Combining (69) and (72) gives

$$\lambda_n = \langle \mathcal{K}\phi_n, \phi_n\rangle_{L^2[0,1]} = \int_0^1 (\mathcal{R}\phi_n)^2(u)du, \quad \forall n \in \mathcal{N}. \tag{74}$$

Similarly,

$$\int_0^1 (\mathcal{R}\psi_\xi)^2(u)du = \langle \mathcal{K}\psi_\xi, \psi_\xi\rangle_{L^2[0,1]} = \sum_{n\in\mathcal{N}} \lambda_n \langle \psi_\xi, \phi_n\rangle_{L^2[0,1]}^2 = 0, \quad \forall \xi \in \Xi, \tag{75}$$

which gives

$$(\mathcal{R}\psi_\xi)(u) = \int_0^1 R(u,v)\psi_\xi(v)dv = 0, \quad a.e. \ u \in [0,1]. \tag{76}$$

Notice that $R_u(v) := R(u,v) \in C^\alpha[0,1] \subset L^2[0,1]$ for any $u \in [0,1]$. By (70) and (76), we get

$$\|R_u\|_{L^2[0,1]}^2 = \Bigg\langle \sum_{n\in\mathcal{N}} \langle R_u, \phi_n\rangle_{L^2[0,1]}\phi_n + \sum_{\xi\in\Xi} \langle R_u, \psi_\xi\rangle_{L^2[0,1]}\psi_\xi,$$

$$\sum_{n\in\mathcal{N}} \langle R_u, \phi_n\rangle_{L^2[0,1]}\phi_n + \sum_{\xi\in\Xi} \langle R_u, \psi_\xi\rangle_{L^2[0,1]}\psi_\xi \Bigg\rangle_{L^2[0,1]}$$

$$= \sum_{n\in\mathcal{N}} \langle R_u, \phi_n\rangle_{L^2[0,1]}^2 + \sum_{\xi\in\Xi} \langle R_u, \psi_\xi\rangle_{L^2[0,1]}^2$$

$$= \sum_{n\in\mathcal{N}} (\mathcal{R}\phi_n)^2(u), \quad a.e. \ u \in [0,1]. \tag{77}$$

Hence for any $N \in \mathbb{N}$, we have

$$\left\| R_u - \sum_{n=1}^N \langle R_u, \phi_n\rangle_{L^2[0,1]}\phi_n \right\|_{L^2[0,1]}^2 = \|R_u\|_{L^2[0,1]}^2 - \sum_{n=1}^N \langle R_u, \phi_n\rangle_{L^2[0,1]}^2 \tag{78}$$

$$= \sum_{n\in\mathcal{N}} (\mathcal{R}\phi_n)^2(u) - \sum_{n=1}^N (\mathcal{R}\phi_n)^2(u)$$

$$= \sum_{n=N+1}^{N_0} (\mathcal{R}\phi_n)^2(u), \tag{79}$$

where the summation is zero by convention if the subscript is larger than the superscript. This by (74) implies

$$\int_0^1 \left\| R(u,v) - \sum_{n=1}^N \langle R(u,v), \phi_n(v)\rangle_{L^2[0,1]} \phi_n(v) \right\|_{L^2[0,1]}^2 du = \sum_{n=N+1}^{N_0} \lambda_n. \tag{80}$$

Here, the equality holds as a consequence of Beppo Levi's monotone convergence lemma and Lebesgue's dominated convergence theorem, and one has $\sum_{n\in\mathcal{N}} \lambda_n < +\infty$. In fact, for

$N_0 = +\infty$, let $S_n = \sum_{k=1}^{n} \lambda_k$, we get $S_n$ increasing (since $\lambda_k \geq 0$ for all $k \in \mathcal{N}$). By (74) and (78), we have

$$S_n = \sum_{k=1}^{n} \int_0^1 (\mathcal{R}\phi_k)^2(u)du = \int_0^1 \sum_{k=1}^{n} \langle R_u, \phi_k \rangle_{L^2[0,1]}^2 du \leq \int_0^1 \|R_u\|_{L^2[0,1]}^2 du = \|R\|_{L^2([0,1]^2)}^2,$$ (81)

which gives that $S_n$ converges as $n \to \infty$. The proof is completed. $\qquad\square$

**Remark D.1.** *Let $\varphi_n(u) := \langle R(u,v), \phi_n(v) \rangle_{L^2[0,1]}$, then we have the POD estimate $R(u,v) \approx \sum_n \varphi_n(u)\phi_n(v)$, where the error is characterised by the tail sum of eigenvalues of the POD operator.*

Recall the POD operator defined in (61). We similarly define

$$\tilde{\mathcal{K}}: \ \phi(v) \mapsto \int_0^1 \int_0^1 \tilde{R}(u,v)\phi(v)dv \cdot \tilde{R}(u,v)du, \quad \phi(v) \in L^2[0,1],$$ (82)

where $\tilde{R}$ is defined as (55), i.e. the approximator constructed in the general approximation theorem before. Obviously, as a polynomial, $\tilde{R} \in L^\infty([0,1]^2)$. Hence, by Proposition D.1, $\tilde{\mathcal{K}}$ is also linear, bounded, compact, self-adjoint and non-negative. In addition, according to Theorem D.1, we have the following POD estimate

$$\int_0^1 \left\| \tilde{R}(u,v) - \sum_{n=1}^{N} \left\langle \tilde{R}(u,v), \tilde{\phi}_n(v) \right\rangle_{L^2[0,1]} \tilde{\phi}_n(v) \right\|_{L^2[0,1]}^2 du = \sum_{n=N+1}^{\tilde{N}_0} \tilde{\lambda}_n,$$ (83)

where $\{\tilde{\lambda}_n\}_{n=1}^{\tilde{N}_0}$ are eigenvalues of $\tilde{\mathcal{K}}$ satisfying $\tilde{\lambda}_1 \geq \tilde{\lambda}_2 \geq \cdots \geq \tilde{\lambda}_n \geq \cdots \geq 0$, and $\{\tilde{\phi}_n\}_{n=1}^{\tilde{N}_0} \subset L^2[0,1]$ are the corresponding orthonormal eigenfunctions, i.e. $\tilde{\mathcal{K}}\tilde{\phi}_n = \tilde{\lambda}_n\tilde{\phi}_n$, $\tilde{\lambda}_n > 0$ for $n \in \{1, 2, \cdots, \tilde{N}_0\}$.

**Lemma D.1.** *$\tilde{\mathcal{K}}$ is a finite-rank operator. That is, $\tilde{N}_0 \leq \bar{m} = \min\{m_E, m_D\} < \infty$.*

*Proof.* Let $\tilde{\varphi}_n(u) := \left\langle \tilde{R}(u,v), \tilde{\phi}_n(v) \right\rangle_{L^2[0,1]}$. We first show that both $\tilde{\varphi}_n(u)$ and $\tilde{\phi}_n(v)$ are polynomials. In fact, since $\tilde{R}(u,v) = \sum_{i=1}^{\bar{m}} \sum_{j=1}^{\bar{m}} \tilde{r}_{ij} u^i v^j$, we have

$$\left| \frac{\partial^k}{\partial u^k} \tilde{R}(u,v) \right| = \left| \sum_{i=k}^{\bar{m}} \sum_{j=1}^{\bar{m}} \tilde{r}_{ij} \frac{i!}{(i-k)!} u^{i-k} v^j \right|$$

$$\leq \sum_{i=k}^{\bar{m}} \sum_{j=1}^{\bar{m}} |\tilde{r}_{ij}| \frac{i!}{(i-k)!} \triangleq C_1(k,\bar{m}), \quad k = 1, 2, \cdots,$$ (84)

with the convention that the summation is zero if the subscript is larger than the superscript, i.e. $C_1(k,\bar{m}) = 0$ for any $k > \bar{m}$. Let $C_1(\bar{m}) := \max\{\|\tilde{R}\|_{L^\infty([0,1]^2)}, \max_{1 \leq k \leq \bar{m}} C_1(k,\bar{m})\}$, then we have

$$\left| \frac{\partial^k}{\partial u^k} \tilde{R}(u,v)\tilde{\phi}_n(v) \right| \leq C_1(\bar{m})|\tilde{\phi}_n(v)| \in L^2[0,1] \subset L^1[0,1], \quad k = 0, 1, \cdots.$$ (85)

According to Lebesgue's dominated convergence theorem, we get by induction that

$$\frac{d^k}{du^k} \tilde{\varphi}_n(u) = \frac{d^k}{du^k} \int_0^1 \tilde{R}(u,v)\tilde{\phi}_n(v)dv = \int_0^1 \frac{\partial^k}{\partial u^k} \tilde{R}(u,v)\tilde{\phi}_n(v)dv, \quad k = 0, 1, \cdots.$$ (86)

Similarly, we get

$$\left| \frac{\partial^k}{\partial v^k} \tilde{R}(u,v) \right| = \left| \sum_{i=1}^{\bar{m}} \sum_{j=k}^{\bar{m}} \tilde{r}_{ij} u^i \frac{j!}{(j-k)!} v^{j-k} \right|$$

$$\leq \sum_{i=1}^{\bar{m}} \sum_{j=k}^{\bar{m}} |\tilde{r}_{ij}| \frac{j!}{(j-k)!} \triangleq C_2(k,\bar{m}), \quad k = 1, 2, \cdots,$$ (87)

and $C_2(k, \bar{m}) = 0$ for any $k > \bar{m}$. Let $C_2(\bar{m}) := \max\{\|\tilde{R}\|_{L^\infty([0,1]^2)}, \max_{1 \le k \le \bar{m}} C_2(k, \bar{m})\}$, then for $k = 0, 1, \cdots$, we have

$$\left| \frac{\partial^k}{\partial v^k} \tilde{R}(u, v) \int_0^1 \tilde{R}(u, v) \tilde{\phi}_n(v) dv \right| \le C_2(\bar{m}) \|\tilde{R}\|_{L^\infty([0,1]^2)} \|\tilde{\phi}_n\|_{L^1[0,1]}$$
$$\le C_2^2(\bar{m}) \|\tilde{\phi}_n\|_{L^2[0,1]} = C_2^2(\bar{m}) \subset L^1[0,1]. \tag{88}$$

According to Lebesgue's dominated convergence theorem, we get by induction that

$$\frac{d^k}{dv^k} \tilde{\phi}_n(v) = \frac{1}{\tilde{\lambda}_n} \int_0^1 \frac{\partial^k}{\partial v^k} \tilde{R}(u, v) \int_0^1 \tilde{R}(u, v) \tilde{\phi}_n(v) dv du, \quad k = 0, 1, \cdots. \tag{89}$$

That is, $\tilde{\varphi}_n, \tilde{\phi}_n \in C^\infty[0,1]$ for any $n = 1, 2, \cdots, \tilde{N}_0$. Since $\tilde{R}(u, 0) = \tilde{R}(0, v) = 0$ for $u, v \in [0,1]$, we get $\tilde{\varphi}_n(0) = \tilde{\phi}_n(0) = 0$. Furthermore, we have $\frac{d^k}{du^k} \tilde{\varphi}_n(u) = \frac{d^k}{dv^k} \tilde{\phi}_n(v) = 0$ for $k > \bar{m}$, hence $\tilde{\varphi}_n, \tilde{\phi}_n \in \mathcal{P}_{\bar{m}}$. Since $\{\tilde{\phi}_n\}_{n=1}^{\tilde{N}_0} \subset L^2[0,1]$ are orthonormal, we must have $\tilde{N}_0 \le \bar{m} < \infty$. [4] The proof is completed. $\qquad\square$

## D.2 APPROXIMATION RATES

**Perturbation of eigenvalues.** First, we need to bound the gap between the eigenvalues $\{\tilde{\lambda}_n\}_{n=1}^{\tilde{N}_0}$ and $\{\lambda_n\}_{n=1}^{N_0}$ (corresponding to the function $R$ defined in (44)). The following theorem is necessary.

**Theorem D.2** (Courant–Fischer–Weyl min-max principle; Lax (2002) (Chapter 28, Theorem 4)). *Let $\mathcal{B}$ be a compact, self-adjoint operator on a Hilbert space $\mathcal{H}$, whose positive eigenvalues are listed in a decreasing order $\mu_1 \ge \mu_2 \ge \cdots \ge \mu_k \ge \cdots > 0$. Then*

$$\max_{\mathcal{S}_k} \min_{x \in \mathcal{S}_k, \|x\|_{\mathcal{H}} = 1} \langle \mathcal{B}x, x \rangle_{\mathcal{H}} = \mu_k, \tag{90}$$

*where $\mathcal{S}_k \subset \mathcal{H}$ is any $k$-dimensional linear subspace.*

Based on it, we have the following lemma to characterise the perturbation of singular values.

**Lemma D.2.** *For any $R_1, R_2 \in L^\infty([0,1]^2)$, we have the estimate*

$$\left| \sqrt{\lambda_k^{R_1}} - \sqrt{\lambda_k^{R_2}} \right| \le \|R_1 - R_2\|_{L^2([0,1]^2)}. \tag{91}$$

*Proof.* According to Theorem D.2 and by (69), we have

$$\lambda_k^R = \max_{\mathcal{S}_k} \min_{\phi \in \mathcal{S}_k, \|\phi\|_{L^2[0,1]} = 1} \langle \mathcal{K}_R \phi, \phi \rangle_{L^2[0,1]} = \max_{\mathcal{S}_k} \min_{\phi \in \mathcal{S}_k, \|\phi\|_{L^2[0,1]} = 1} \|\mathcal{R}_R \phi\|_{L^2[0,1]}^2. \tag{92}$$

Note that $\mathcal{R}_{R_1} - \mathcal{R}_{R_2} = \mathcal{R}_{R_1 - R_2}$ and by (64), we have

$$\sqrt{\lambda_k^{R_1}} = \max_{\mathcal{S}_k} \min_{\phi \in \mathcal{S}_k, \|\phi\|_{L^2[0,1]} = 1} \|\mathcal{R}_{R_1} \phi\|_{L^2[0,1]}$$
$$\le \max_{\mathcal{S}_k} \min_{\phi \in \mathcal{S}_k, \|\phi\|_{L^2[0,1]} = 1} \left( \|(\mathcal{R}_{R_1} - \mathcal{R}_{R_2})\phi\|_{L^2[0,1]} + \|\mathcal{R}_{R_2} \phi\|_{L^2[0,1]} \right)$$
$$\le \max_{\mathcal{S}_k} \min_{\phi \in \mathcal{S}_k, \|\phi\|_{L^2[0,1]} = 1} \left( \|\mathcal{R}_{R_1 - R_2}\| + \|\mathcal{R}_{R_2} \phi\|_{L^2[0,1]} \right)$$
$$\le \max_{\mathcal{S}_k} \min_{\phi \in \mathcal{S}_k, \|\phi\|_{L^2[0,1]} = 1} \|\mathcal{R}_{R_2} \phi\|_{L^2[0,1]} + \|R_1 - R_2\|_{L^2([0,1]^2)}$$
$$= \sqrt{\lambda_k^{R_2}} + \|R_1 - R_2\|_{L^2([0,1]^2)}, \tag{93}$$

---

[4] In fact, for any $p \in \mathcal{P}_{\bar{m}}$ with $p(0) = 0$, we have $p \in \text{span}\{v, v^2, \cdots, v^{\bar{m}}\}$. Through a standard Schmidt-orthogonalization, we can get $e_k(v) \in \mathcal{P}_k$, $k = 1, 2, \cdots, \bar{m}$, such that $\langle e_i, e_j \rangle_{L^2[0,1]} = \delta_{ij}$, and $p \in \text{span}\{e_1(v), e_2(v), \cdots, e_{\bar{m}}(v)\}$. That is, if $\langle p, q \rangle_{L^2[0,1]} = 0$ for some $p, q \in \mathcal{P}_{\bar{m}}$ with $p(0) = 0$, $q(0) = 0$, then their coordinates under the basis $\{e_k\}_{k=1}^{\bar{m}}$ are orthogonal. Hence, the $\tilde{N}_0$ orthogonal $\bar{m}$-dimensional coordinates here leads to at most $\bar{m}$ non-zeros.

and similarly,

$$
\begin{aligned}
\sqrt{\lambda_k^{R_2}} &= \max_{\mathcal{S}_k} \min_{\phi \in \mathcal{S}_k,\, \|\phi\|_{L^2[0,1]}=1} \|\mathcal{R}_{R_2}\phi\|_{L^2[0,1]} \\
&\le \max_{\mathcal{S}_k} \min_{\phi \in \mathcal{S}_k,\, \|\phi\|_{L^2[0,1]}=1} \left( \|(\mathcal{R}_{R_2} - \mathcal{R}_{R_1})\phi\|_{L^2[0,1]} + \|\mathcal{R}_{R_1}\phi\|_{L^2[0,1]} \right) \\
&\le \max_{\mathcal{S}_k} \min_{\phi \in \mathcal{S}_k,\, \|\phi\|_{L^2[0,1]}=1} \left( \|\mathcal{R}_{R_2-R_1}\| + \|\mathcal{R}_{R_1}\phi\|_{L^2[0,1]} \right) \\
&\le \max_{\mathcal{S}_k} \min_{\phi \in \mathcal{S}_k,\, \|\phi\|_{L^2[0,1]}=1} \|\mathcal{R}_{R_1}\phi\|_{L^2[0,1]} + \|R_2 - R_1\|_{L^2([0,1]^2)} \\
&= \sqrt{\lambda_k^{R_1}} + \|R_1 - R_2\|_{L^2([0,1]^2)},
\end{aligned}
\tag{94}
$$

which completes the proof. $\qquad\square$

**Proofs.** Now we are ready to derive the final estimate.

*Proof of Theorem 4.3.* By Lemma D.2 and (54), we get

$$
\left| \sqrt{\lambda_k} - \sqrt{\tilde{\lambda}_k} \right| \le \|R - \tilde{R}\|_{L^2([0,1]^2)} \le \|R - \tilde{R}\|_{L^\infty([0,1]^2)} \le \frac{C(\alpha)\gamma}{\bar{m}^\alpha}.
\tag{95}
$$

Combining (54), (83) and (95) gives that

$$
\begin{aligned}
&\frac{1}{c_0^2} \left\| \rho(t,s) - \sum_{n=1}^{N} e^{-\frac{t}{c_0}} \tilde{\varphi}_n(e^{-\frac{t}{c_0}}) \cdot e^{-\frac{s}{c_0}} \tilde{\phi}_n(e^{-\frac{s}{c_0}}) \right\|_{L^1([0,\infty)^2)} \\
&= \left\| R(u,v) - \sum_{n=1}^{N} \tilde{\varphi}_n(u)\tilde{\phi}_n(v) \right\|_{L^1([0,1]^2)} \\
&\le \left\| R(u,v) - \tilde{R}(u,v) \right\|_{L^1([0,1]^2)} + \left\| \tilde{R}(u,v) - \sum_{n=1}^{N} \tilde{\varphi}_n(u)\tilde{\phi}_n(v) \right\|_{L^1([0,1]^2)} \\
&\le \left\| R(u,v) - \tilde{R}(u,v) \right\|_{L^\infty([0,1]^2)} + \left\| \tilde{R}(u,v) - \sum_{n=1}^{N} \tilde{\varphi}_n(u)\tilde{\phi}_n(v) \right\|_{L^2([0,1]^2)} \\
&\le \frac{C(\alpha)\gamma}{\bar{m}^\alpha} + \sqrt{\sum_{n=N+1}^{\tilde{N}_0} \tilde{\lambda}_n} \le \frac{C(\alpha)\gamma}{\bar{m}^\alpha} + \sqrt{\sum_{n=N+1}^{\tilde{N}_0} |\tilde{\lambda}_n - \lambda_n| + \sum_{n=N+1}^{\tilde{N}_0} \lambda_n} \\
&\le \frac{C(\alpha)\gamma}{\bar{m}^\alpha} + \sqrt{\sum_{n=N+1}^{\tilde{N}_0} \lambda_n} + \sqrt{\sum_{n=N+1}^{\tilde{N}_0} \left| \sqrt{\tilde{\lambda}_n} - \sqrt{\lambda_n} \right| \left| \sqrt{\tilde{\lambda}_n} + \sqrt{\lambda_n} \right|} \\
&\le \frac{C(\alpha)\gamma}{\bar{m}^\alpha} + \sqrt{\sum_{n=N+1}^{\tilde{N}_0} \lambda_n} + \sqrt{\sum_{n=N+1}^{\tilde{N}_0} \left| \sqrt{\tilde{\lambda}_n} - \sqrt{\lambda_n} \right|^2} + \sqrt{2} \sqrt{\sum_{n=N+1}^{\tilde{N}_0} \sqrt{\lambda_n} \left| \sqrt{\tilde{\lambda}_n} - \sqrt{\lambda_n} \right|} \\
&\lesssim C(\alpha)\gamma \left\{ \left( 1 + \sqrt{\tilde{N}_0 - N} \right) \cdot \frac{1}{\bar{m}^\alpha} + \sqrt{\sum_{n=N+1}^{\tilde{N}_0} \lambda_n} + \sqrt{\sum_{n=N+1}^{\tilde{N}_0} \sqrt{\lambda_n} \cdot \frac{1}{\bar{m}^{\alpha/2}}} \right\},
\end{aligned}
\tag{96}
$$

where $\lesssim$ hides universal positive constants.

For the corresponding parameters, recall that $\tilde{\varphi}_n, \tilde{\phi}_n \in \mathcal{P}_{\bar{m}}$ with $\tilde{\varphi}_n(0) = \tilde{\phi}_n(0) = 0$, we can write

$$
\tilde{\varphi}_n(u) = \sum_{i=1}^{\bar{m}} P_{in} u^i, \quad \tilde{\phi}_n(v) = \sum_{j=1}^{\bar{m}} Q_{nj} v^j
\tag{97}
$$

for any $n = 1, 2, \cdots, \tilde{N}_0$. Let

$$c = \mathbf{1}_{\bar{m}}, \quad u = \mathbf{1}_{\bar{m}}, \tag{98}$$

$$V = -\text{diag}(2, 3, \cdots, \bar{m} + 1)/c_0, \quad W = -\text{diag}(2, 3, \cdots, \bar{m} + 1)/c_0, \tag{99}$$

$$M = PQ \text{ with } P = [P_{in}] \in \mathbb{R}^{\bar{m} \times N}, \ Q = [Q_{nj}] \in \mathbb{R}^{N \times \bar{m}}, \tag{100}$$

then we have

$$c^\top e^{Vt} M e^{Ws} u = c^\top e^{Vt} P \cdot Q e^{Ws} u = \sum_{n=1}^{N} \sum_{i=1}^{\bar{m}} e^{-\frac{i+1}{c_0} t} P_{in} \cdot \sum_{j=1}^{\bar{m}} Q_{nj} e^{-\frac{j+1}{c_0} s}$$

$$= \sum_{n=1}^{N} e^{-\frac{t}{c_0}} \tilde{\varphi}_n(e^{-\frac{t}{c_0}}) \cdot e^{-\frac{s}{c_0}} \tilde{\phi}_n(e^{-\frac{s}{c_0}}). \tag{101}$$

Plugging this into (96) gives

$$\|\boldsymbol{H} - \widehat{\boldsymbol{H}}\| \le \|\rho - \hat{\rho}\|_{L^1([0,\infty)^2)} \tag{102}$$

$$= \left\| \rho(t, s) - c^\top e^{Vt} M e^{Ws} u \right\|_{L^1([0,\infty)^2)}$$

$$\lesssim \frac{C(\alpha)\gamma}{\beta^2} \left\{ \left(1 + \sqrt{\tilde{N}_0 - N}\right) \cdot \frac{1}{\bar{m}^\alpha} + \sqrt{\sum_{n=N+1}^{\tilde{N}_0} \lambda_n} + \sqrt{\sum_{n=N+1}^{\tilde{N}_0} \sqrt{\lambda_n}} \cdot \frac{1}{\bar{m}^{\alpha/2}} \right\}, \tag{103}$$

and the number of trainable parameters is $2N\bar{m}$. Together with Lemma D.1, the proof is completed. $\quad\square$

**Extension to multi-dimensional inputs.** The above results can be naturally extended to the general case where a $d$-dimensional input is given ($\forall d \in \mathbb{N}_+$). In fact, let

$$\left\| \rho_i(t, s) - c^\top e^{Vt} M_i e^{Ws} u \right\|_{L^1([0,\infty)^2)} \lesssim \epsilon, \quad i = 1, 2, \cdots, d, \tag{104}$$

for some $0 < \epsilon \ll 1$, then we take

$$M = (M_1, M_2, \cdots, M_d) \in \mathbb{R}^{\bar{m} \times d\bar{m}}, \tag{105}$$

$$W = \text{diag}(W, W, \cdots, W) \in \mathbb{R}^{d\bar{m} \times d\bar{m}}, \quad U = \text{diag}(u, u, \cdots, u) \in \mathbb{R}^{d\bar{m} \times d}, \tag{106}$$

and have

$$c^\top e^{Vt} M e^{Ws} U = c^\top e^{Vt} \cdot (M_1, M_2, \cdots, M_d) \cdot \text{diag}(e^{Ws}, e^{Ws}, \cdots, e^{Ws}) \cdot \text{diag}(u, u, \cdots, u)$$

$$= (c^\top e^{Vt} M_1, c^\top e^{Vt} M_2, \cdots, c^\top e^{Vt} M_d) \cdot \text{diag}(e^{Ws} u, e^{Ws} u, \cdots, e^{Ws} u)$$

$$= (c^\top e^{Vt} M_1 e^{Ws} u, c^\top e^{Vt} M_2 e^{Ws} u, \cdots, c^\top e^{Vt} M_d e^{Ws} u). \tag{107}$$

If the form $PQ(= M)$ is required, it is sufficient to take $M_i = P_i Q_i$, $i = 1, 2, \cdots, d$, and

$$P = (P_1, P_2, \cdots, P_d) \in \mathbb{R}^{\bar{m} \times dN}, \ Q = \text{diag}(Q_1, Q_2, \cdots, Q_d) \in \mathbb{R}^{dN \times d\bar{m}}. \tag{108}$$

Therefore, we obtain

$$\sum_{i=1}^{d} \left\| \rho_i(t, s) - \left[c^\top e^{Vt} M e^{Ws} U\right]_i \right\|_{L^1([0,\infty)^2)} \lesssim d\epsilon, \tag{109}$$

with the number of parameters increased by $d$-times compared to the corresponding one-dimensional setting.

### D.3 Case analysis

**Bounds under different cases.** Now we make the comparison between (102) and (58). Recall that $\{\lambda_n\}_{n=1}^{N_0}$ is a positive decreased sequence (with $\lim_{n\to\infty} \lambda_n = 0$ and $\sum_{n=1}^{\infty} \lambda_n \le \|R\|_{L^2([0,1]^2)}^2$ by (81), if $N_0 = +\infty$), and $\tilde{N}_0 \le \bar{m}$, we have the following cases.

- if $\tilde{N}_0 = o(\bar{m})$, we set $N = \tilde{N}_0$ in (102) and get the same bound as (58), but the number of parameters is only $\mathcal{O}(\tilde{N}_0\bar{m}) = o(\bar{m}^2)$;

- if $\tilde{N}_0 = \mathcal{O}(\bar{m})$, then (102) implies that

$$\left\| \rho(t,s) - c^\top e^{Vt} M e^{Ws} u \right\|_{L^1([0,\infty)^2)}$$

$$\lesssim \frac{C(\alpha)\gamma}{\beta^2} \left\{ \left(1 + \sqrt{\bar{m} - N}\right) \cdot \frac{1}{\bar{m}^\alpha} + \sqrt{\sum_{n=N+1}^{\bar{m}} \lambda_n} + \sqrt{\sum_{n=N+1}^{\bar{m}} \sqrt{\lambda_n}} \cdot \frac{1}{\bar{m}^{\alpha/2}} \right\}$$

$$\lesssim \frac{C(\alpha)\gamma}{\beta^2} \left\{ \frac{1}{\bar{m}^{\alpha-\frac{1}{2}}} + \sqrt{\sum_{n=N+1}^{\bar{m}} \lambda_n} + \sqrt{\sum_{n=N+1}^{\bar{m}} \sqrt{\lambda_n}} \cdot \frac{1}{\bar{m}^{\alpha/2}} \right\}. \tag{110}$$

We are supposed to require that

$$\frac{1}{\bar{m}^{\alpha-\frac{1}{2}}} \gtrsim \max \left\{ \sqrt{\sum_{n=N+1}^{\bar{m}} \lambda_n}, \sqrt{\sum_{n=N+1}^{\bar{m}} \sqrt{\lambda_n}} \cdot \frac{1}{\bar{m}^{\alpha/2}} \right\}$$

$$\Leftrightarrow \sum_{n=N+1}^{\bar{m}} \lambda_n \lesssim \frac{1}{\bar{m}^{2\alpha-1}}, \quad \sum_{n=N+1}^{\bar{m}} \sqrt{\lambda_n} \lesssim \frac{1}{\bar{m}^{\alpha-1}}. \tag{111}$$

We give the following typical examples to illustrate sufficient conditions to guarantee (111).

- If $\lambda_n = \mathcal{O}(n^{-r})$ with $r > 2\alpha + 1 \geq 3$ ($\alpha \in \mathbb{N}_+$), since

$$\sum_{n=N+1}^{\bar{m}} n^{-r} \leq \int_N^{\bar{m}} x^{-r} dx = \frac{1}{r-1} \left( \frac{1}{N^{r-1}} - \frac{1}{\bar{m}^{r-1}} \right), \quad \forall r > 1, \tag{112}$$

we get by (111) that

$$\sum_{n=N+1}^{\bar{m}} \lambda_n \lesssim \sum_{n=N+1}^{\bar{m}} n^{-r} \lesssim \frac{1}{N^{r-1}} \lesssim \frac{1}{\bar{m}^{2\alpha-1}} \Leftrightarrow N \gtrsim \bar{m}^{\frac{2\alpha-1}{r-1}}, \tag{113}$$

$$\sum_{n=N+1}^{\bar{m}} \sqrt{\lambda_n} \lesssim \sum_{n=N+1}^{\bar{m}} n^{-\frac{r}{2}} \lesssim \frac{1}{N^{\frac{r}{2}-1}} \lesssim \frac{1}{\bar{m}^{\alpha-1}} \Leftrightarrow N \gtrsim \bar{m}^{\frac{\alpha-1}{\frac{r}{2}-1}}. \tag{114}$$

Assume that $N \sim \bar{m}^\delta$ with $\delta \in [0,1)$, then by (113) and (114), we require $\delta \geq \max\{\frac{2\alpha-1}{r-1}, \frac{\alpha-1}{\frac{r}{2}-1}\}$, i.e.

$$r \geq \max \left\{ \frac{2\alpha-1}{\delta} + 1, 2\left(\frac{\alpha-1}{\delta} + 1\right) \right\} = \frac{2\alpha-1}{\delta} + 1. \tag{115}$$

Meanwhile, the POD-estimate (110) achieves an accuracy scaling like $(1/\bar{m})^{\alpha-\frac{1}{2}}$ with $\mathcal{O}(\bar{m}^{1+\delta})$ parameters, while under the same capacity, the accuracy of (58) scales like $(1/\bar{m})^{\frac{\alpha(1+\delta)}{2}}$. The former beats the latter if $\alpha - \frac{1}{2} > \frac{\alpha(1+\delta)}{2}$, i.e. $\delta < 1 - 1/\alpha$ ($\alpha \geq 2$). When $\delta = 1 - 1/\alpha$, (115) becomes $r \geq \max\{\frac{2\alpha^2-1}{\alpha-1}, 2(\alpha+1)\} = \frac{2\alpha^2-1}{\alpha-1}$. That is to say, $r^* := \frac{2\alpha^2-1}{\alpha-1}$ can be viewed as an upper bound of the critical point where the two estimates are comparable. When $r > r^*$, the POD-estimate outperforms the non-POD-estimate, and this effect gets more notable with $r$ increasing. In fact, when $r > r^*$, we take $N = \bar{m}^{\frac{2\alpha-1}{r-1}} > \bar{m}^{\frac{\alpha-1}{\frac{r}{2}-1}}$ (hence satisfying (113) and (114)), which gives an $\mathcal{O}((1/\bar{m})^{\alpha-\frac{1}{2}})$ approximation error with $\mathcal{O}(\bar{m}^{1+\frac{2\alpha-1}{r-1}})$ parameters for the POD-estimate, while $\mathcal{O}(\bar{m}^{2-\frac{1}{\alpha}})$ trainable parameters are needed to achieve the same accuracy using the non-POD-estimate. Since $\frac{2\alpha-1}{r-1} < 1 - \frac{1}{\alpha}$, we get that the POD-estimate outperforms the non-POD-estimate. When $r \to +\infty$, the number of trainable parameters for the POD-estimate is $\mathcal{O}(\bar{m})$, much better than the non-POD-estimate.

**Remark D.2.** *Let*

$$K(v, w) := \int_0^1 R(u, v) R(u, w) du, \quad v, w \in [0, 1], \tag{116}$$

*we get*

$$(\mathcal{K}\phi)(w) = \int_0^1 K(v, w)\phi(v) dv, \quad \phi \in L^2[0, 1]. \tag{117}$$

*Recall that $R \in C^\alpha([0,1]^2) \subset L^\infty([0,1]^2)$, we get $K \in L^\infty([0,1]^2) \subset L^2([0,1]^2)$, and hence $\mathcal{K}$ can be also viewed as a Hilbert-Schmidt integral operator with the kernel $K$, where $K$ is obviously symmetric as $K(v, w) = K(w, v)$, and positive definite since $0 \leq \langle \mathcal{K}\phi, \phi \rangle_{L^2[0,1]} = \int_0^1 \int_0^1 K(v, w)\phi(v)\phi(w) dv dw$, $\phi \in L^2[0, 1]$ by (69). Since the derivatives of $R$ (up to $\alpha$-order) are continuous on $[0, 1]^2$ (hence bounded and integrable), we get $K \in C^{\alpha,\alpha}([0,1]^2)$ ($\alpha$-differentiable for both arguments). According to Chang & Ha (1999) (Theorem 1), we have*

$$\sum_{n=N+2\alpha+1}^\infty \lambda_n \lesssim N^{-\alpha}, \quad \forall N \in \mathbb{N}_+ \Rightarrow \lambda_n = \mathcal{O}(n^{-(\alpha+1)}), \quad n \to \infty. \tag{118}$$

*That is to say, this general setting (only assume smoothness of the kernel) can not guarantee the sufficient condition provided here ($\lambda_n = \mathcal{O}(n^{-r})$ with $r > \frac{2\alpha^2 - 1}{\alpha - 1}$), i.e. the point that the POD-estimate outperforms the non-POD-estimate requires a faster decay of the eigenvalues.*

- If $\lambda_n = \mathcal{O}(e^{-\omega n})$ with $\omega > 0$, we get by (111) that

$$\sum_{n=N+1}^{\bar{m}} \lambda_n \lesssim \sum_{n=N+1}^\infty e^{-\omega n} = \frac{e^{-\omega(N+1)}}{1 - e^{-\omega}} \lesssim \frac{1}{\bar{m}^{2\alpha-1}} \Leftrightarrow N \gtrsim \frac{1}{\omega} \ln\left(\frac{\bar{m}^{2\alpha-1}}{1 - e^{-\omega}}\right) - 1, \tag{119}$$

$$\sum_{n=N+1}^{\bar{m}} \sqrt{\lambda_n} \lesssim \sum_{n=N+1}^\infty e^{-\frac{\omega}{2}n} = \frac{e^{-\frac{\omega}{2}(N+1)}}{1 - e^{-\frac{\omega}{2}}} \lesssim \frac{1}{\bar{m}^{\alpha-1}} \Leftrightarrow N \gtrsim \frac{2}{\omega} \ln\left(\frac{\bar{m}^{\alpha-1}}{1 - e^{-\frac{\omega}{2}}}\right) - 1. \tag{120}$$

  That is to say, for any $\alpha \in \mathbb{N}_+$, $\omega > 0$, we have $N \sim (2\alpha - 1) \ln \bar{m}$. This implies an $\mathcal{O}((1/\bar{m})^{\alpha - \frac{1}{2}})$ approximation error with $\mathcal{O}(\bar{m} \ln \bar{m})$ parameters for the POD-estimate, while $\mathcal{O}(\bar{m}^{2-\frac{1}{\alpha}})$ parameters are needed to achieve the same accuracy using the non-POD-estimate.

- If $N_0 < +\infty$ (i.e. $\mathcal{K}$ is a finite rank operator by (71)), one can just take $N = N_0$ and get by (110) an $\mathcal{O}((1/\bar{m})^{\alpha - \frac{1}{2}})$ approximation error. For $\bar{m} \in \mathbb{N}_+$ sufficiently large such that $\bar{m} \sim N_0^\kappa$ for some $\kappa \gg 1$, the number of trainable parameters for the POD-estimate is $\mathcal{O}(N\bar{m}) = \mathcal{O}(N_0^{\kappa+1})$, while $\mathcal{O}\left(N_0^{\kappa(2-\frac{1}{\alpha})}\right)$ parameters are needed to achieve the same accuracy using the non-POD-estimate. Obviously, if $\alpha \geq 2$, we get $\kappa(2 - \frac{1}{\alpha}) \geq \frac{3}{2}\kappa \gg \kappa + 1$.

**Eigenvalue approximation.** One can apply (91) in Lemma D.2 to estimate the eigenvalues $\{\lambda_n\}_{n=1}^{N_0}$, where $R_1 = R$ and $R_2$ is taken as some approximator of $R$, say $\hat{R}$. Let $\hat{\lambda} := \lambda^{\hat{R}}$, we recall (91) as

$$\left|\sqrt{\lambda_k} - \sqrt{\hat{\lambda}_k}\right| \leq \|R - \hat{R}\|_{L^2([0,1]^2)}. \tag{121}$$

We provide a naive method here. That is, one can take $\hat{R}$ as a piece-wise constant approximation of $R$. Fix any $n \in \mathbb{N}_+$. Let $\mathcal{I}_0 := \{0\}$, $\mathcal{I}_i := (\frac{i-1}{n}, \frac{i}{n}]$ for $i = 1, 2, \cdots, n$, then $\{\mathcal{I}_i \times \mathcal{I}_j\}_{i,j=0}^n$ is the uniform partition over $[0, 1]^2$. Let $\hat{R}(u, v) := \sum_{i,j=0}^n R(\frac{i}{n}, \frac{j}{n}) \mathbf{1}_{\mathcal{I}_i \times \mathcal{I}_j}(u, v)$.

Then for any $\phi \in L^2[0,1]$, if $w \in \mathcal{I}_k$, $k = 0, 1, \cdots, n$, we have

$$
\begin{aligned}
(\mathcal{K}_{\hat{R}}\phi)(w) &= \int_0^1 \int_0^1 \hat{R}(u,v)\phi(v)dv \cdot \hat{R}(u,w)du \\
&= \sum_{i=0}^n \sum_{j=0}^n \sum_{i'=0}^n \sum_{j'=0}^n R\left(\frac{i}{n},\frac{j}{n}\right) R\left(\frac{i'}{n},\frac{j'}{n}\right) \int_0^1 \int_0^1 \mathbf{1}_{\mathcal{I}_i \times \mathcal{I}_j}(u,v)\phi(v)dv \cdot \mathbf{1}_{\mathcal{I}_{i'} \times \mathcal{I}_{j'}}(u,w)du \\
&= \sum_{i=0}^n \sum_{j=0}^n R\left(\frac{i}{n},\frac{j}{n}\right) R\left(\frac{i}{n},\frac{k}{n}\right) \int_{\mathcal{I}_i} \int_{\mathcal{I}_j} \mathbf{1}_{\mathcal{I}_i \times \mathcal{I}_j}(u,v)\phi(v)dv \cdot \mathbf{1}_{\mathcal{I}_i \times \mathcal{I}_k}(u,w)du \\
&= \sum_{i=0}^n \sum_{j=0}^n R\left(\frac{i}{n},\frac{j}{n}\right) R\left(\frac{i}{n},\frac{k}{n}\right) \cdot \frac{1}{n} \int_{\mathcal{I}_j} \phi(v)dv, \quad\quad (122)
\end{aligned}
$$

which is a constant only related to $k$. That is, $\mathcal{K}_{\hat{R}}\phi$ is a piece-wise constant function, i.e. $\mathcal{K}_{\hat{R}}\phi \in \text{span}\{\mathbf{1}_{\mathcal{I}_k}\}_{k=0}^n$, or range$(\mathcal{K}_{\hat{R}}) \subset \text{span}\{\mathbf{1}_{\mathcal{I}_k}\}_{k=0}^n$. Obviously, $\{\mathbf{1}_{\mathcal{I}_k}\}_{k=0}^n$ is an orthogonal set, which implies that the operator $\mathcal{K}_{\hat{R}}$ is of finite rank at most $n+1$. Let $\hat{R}_{ij} := \frac{1}{n}R(\frac{i-1}{n},\frac{j-1}{n})$, $i,j = 1,2,\cdots,n+1$, $\{\sigma_k\}_{k=1}^{n+1}$ be the singular value of $\hat{R} := [\hat{R}_{ij}] \in \mathbb{R}^{(n+1)\times(n+1)}$ and $V \in \mathbb{R}^{(n+1)\times(n+1)}$ with columns as the corresponding right singular vectors. Set $[e_1, e_2, \cdots, e_{n+1}] := [\mathbf{1}_{\mathcal{I}_0}, \mathbf{1}_{\mathcal{I}_1}, \cdots, \mathbf{1}_{\mathcal{I}_n}]V$, then by (122), we get for $l = 1, 2, \cdots, n+1$,

$$
\begin{aligned}
(\mathcal{K}_{\hat{R}}e_l)(w) &= \sum_{i=0}^n \sum_{j=0}^n R\left(\frac{i}{n},\frac{j}{n}\right) R\left(\frac{i}{n},\frac{k}{n}\right) \cdot \frac{1}{n^2} V_{j+1,l} \\
&= \sum_{i=1}^{n+1} \sum_{j=1}^{n+1} \hat{R}_{ij} \hat{R}_{i,k+1} V_{jl} = \left[\hat{R}^\top \hat{R} V_{:,l}\right]_{k+1}, \quad\quad (123)
\end{aligned}
$$

which gives

$$
\mathcal{K}_{\hat{R}}e_l = \sum_{k=0}^n \left[\hat{R}^\top \hat{R} V_{:,l}\right]_{k+1} \mathbf{1}_{\mathcal{I}_k} = \sigma_l^2 \sum_{k=0}^n V_{k+1,l} \mathbf{1}_{\mathcal{I}_k} = \sigma_l^2 e_l, \quad\quad (124)
$$

i.e. the set $\{\sigma_k^2\}_{k=1}^{n+1}$ collects the all the eigenvalues of $\mathcal{K}_{\hat{R}}$, which can be obtained by the SVD of $\hat{R}$.

For the error estimate, it is straightforward to have

$$
\begin{aligned}
&\left\|R(u,v) - \hat{R}(u,v)\right\|_{L^2([0,1]^2)}^2 \\
&= \sum_{i=1}^n \sum_{j=1}^n \int_{\frac{i-1}{n}}^{\frac{i}{n}} \int_{\frac{j-1}{n}}^{\frac{j}{n}} \left|R(u,v) - R\left(\frac{i}{n},\frac{j}{n}\right)\right|^2 dudv \\
&\leq \sum_{i=1}^n \sum_{j=1}^n \int_{\frac{i-1}{n}}^{\frac{i}{n}} \int_{\frac{j-1}{n}}^{\frac{j}{n}} \max_{(u,v)\in[0,1]^2} \|\nabla R(u,v)\|_2^2 \cdot \left\|\left(u-\frac{i}{n}, v-\frac{j}{n}\right)\right\|_2^2 dudv \\
&\leq \sum_{i=1}^n \sum_{j=1}^n \int_{\frac{i-1}{n}}^{\frac{i}{n}} \int_{\frac{j-1}{n}}^{\frac{j}{n}} 2C^2(\alpha)\gamma^2 \cdot \frac{2}{n^2} dudv = \frac{4C^2(\alpha)\gamma^2}{n^2}, \quad\quad (125)
\end{aligned}
$$

hence by (121), $\left|\sqrt{\lambda_k} - \sigma_k\right| \leq \frac{2C(\alpha)\gamma}{n}$ for any $n \in \mathbb{N}_+$.

## E  NUMERICAL SETTINGS

According to Lemma B.1, the target input-output temporal relationship has a Riesz representation form. Under the discrete-time regime, we are supposed to set

$$
H_t(\boldsymbol{x}) = \sum_{s=1}^T \rho(t,s)x_s, \quad\quad (126)
$$

where $T \in \mathbb{N}_+$ is the path length.

### E.1 Settings of Figure 1

For the input, we generate 6 sequences using Gaussian i.i.d. random variables with the path length $T = 30$. Hence, the output (target) is $H_t(\boldsymbol{x}) = \sum_{s=1}^{30} \rho(t, s) x_s$.

The high rank target has the representation

$$\rho^{\text{high}}(t, s) = \begin{cases} \cos(t), & t = s, \\ 0, & t \neq s, \end{cases} \tag{127}$$

while the low rank target has the representation

$$\rho^{\text{low}}(t, s) = \sum_{n=0}^{99} \frac{1}{n+1} \cos(n\pi t) \cos(n\pi s). \tag{128}$$

### E.2 Settings of Figure 2

Consider the target with the following representation

$$\rho(t, s) = e^{-\frac{t}{c_0}} e^{-\frac{s}{c_0}} R(e^{-\frac{t}{c_0}}, e^{-\frac{s}{c_0}}), \tag{129}$$

where

$$R(u, v) = \sum_{n=1}^{\infty} \sigma_n \varphi_n(u) \phi_n(v), \tag{130}$$

with both $\{\varphi_k\}$ and $\{\phi_k\}$ as orthonormal bases. In this way, we construct a target with singular values $\{\sigma_k\}$. Under the discrete-time setting, we are supposed to use the following linear, width-$m$ RNN encoder-decoder

$$\widehat{H}_t(\boldsymbol{x}) = \sum_{s=1}^{\tau} c^{\top} V^t P Q W^{s-1} U x(\tau - s), \tag{131}$$

where $P \in \mathbb{R}^{m \times N}$, $Q \in \mathbb{R}^{N \times m}$ with $m = m_D = m_E$.

Recall that the approximation error is derived as

$$\|\boldsymbol{H} - \widehat{\boldsymbol{H}}\| \lesssim \|R - \hat{R}\|_{L^1([0,1]^2)} \leq \|R - \hat{R}\|_{L^2([0,1]^2)}, \tag{132}$$

where $\hat{R}(u, v) := \sum_{i=1}^{N_0} \tilde{\varphi}_i(u) \tilde{\phi}_i(v) \in \mathcal{P}_m^2$ with $\tilde{\varphi}_n, \tilde{\phi}_n \in \mathcal{P}_m$. In the numerical experiments, we first construct an $R(u, v)$, and then fit it with the polynomial $\hat{R}(u, v)$ using the method of least squares. The norm $\|R - \hat{R}\|_{L^2([0,1]^2)}$ is used to evaluate the approximation error.

In the particular example reported in Figure 2, we set $m = 128$, $\varphi_n(u) = \sqrt{2}\sin(n\pi u)$ and $\phi_n(v) = \sqrt{2}\sin(n\pi v)$. The singular values are taken as: (a) $\sigma_n = \begin{cases} n^{-\frac{1}{8}}, & n \leq N_0 \\ 0, & n > N_0 \end{cases}$; (b) $\sigma_n = \begin{cases} n^{-1}, & n \leq N_0 \\ 0, & n > N_0 \end{cases}$; (c) $\sigma_n = n^{-2}$, with $N_0 = 2, 4, 6, 8$. As an infinite sum, $R$ is constructed under a finite truncation with the first 50 terms.

### E.3 Settings of Figure 3

We perform experiments on nonlinear targets to show that the insight of low rank approximation also holds in the nonlinear case.

**Nonlinear target.** We consider the forced Lorenz 96 system (Lorenz, 1996), which is an important example of reduced order modelling for convection dynamics, with applications in weather forecasting.

Mathematically, the system has $K$ output variables $\{y_k\}$ and $JK$ hidden variables $\{z_{j,k}\}$ with $k = 1, 2, \ldots, K$ and $j = 1, 2, \ldots, J$. The parameters $K, J$ control the number of

variables in the system, and can be viewed as a complexity measure. The input $\{x_k\}$ is an external temporal forcing. The system satisfies the following dynamics

$$\frac{dy_k}{dt} = -y_{k-1}(y_{k-2} - y_{k+1}) - y_k + x_k - \frac{1}{J}\sum_{j=1}^{J} z_{j,k}, \tag{133}$$

$$\frac{dz_{j,k}}{dt} = -z_{j+1,k}(z_{j+2,k} - z_{j-1,k}) - z_{j,k} + y_k, \tag{134}$$

with cyclic indices $y_{k+K} = y_k$, $z_{j,k+K} = z_{j,k}$ and $z_{j+J,k} = z_{j,k}$. Here, we take $\{x_k\}$ as randomly generated input sequences with the path length 64. We have tested for several cases with different parameters: i) $J = 6$ with $K = 1, 5, 10, 20$; ii) $K = 5$ with $J = 5, 15, 25, 100$.

Note that the forced Lorenz 96 system parameterizes a highly nonlinear functional.

**Nonlinear model.** We learn the above system using RNN encoder-decoders with nonlinear activations, i.e.

$$\begin{aligned}
h_s &= \sigma(W_E h_{s-1} + U_E x_s + b_E), & v &= \sigma(Q h_\tau + b_1), \\
g_t &= \sigma(W_D g_{t-1} + b_D), & g_0 &= \sigma(P v + b_2), \\
o_t &= W_O g_t + b_O,
\end{aligned} \tag{135}$$

where $\sigma$ is the element-wise tanh activation. Let $m = 128$ be the hidden dimension, $N = 1, 2, \ldots, 32$ be the size of the coding vector $v$, we have $x_s, o_t, b_O \in \mathbb{R}^K$, $h_s, b_E, b_D, b_2 \in \mathbb{R}^m$, $W_E, W_D \in \mathbb{R}^{m \times m}$, $b_1 \in \mathbb{R}^N$, $W_O \in \mathbb{R}^{m \times K}$, $Q \in \mathbb{R}^{m \times N}$, $P \in \mathbb{R}^{N \times m}$. Note that we construct the model with a fixed hidden dimension $m$ but different $N$, thus only sizes of $Q, P, b_1, b_2$ are varying, while sizes of other parameters remain unchanged.

**Training and initialisation.** We denote the model with the coding vector size $N$ as $\text{EncDec}^{(N)}$. We utilise the Adam optimiser and train from $\text{EncDec}^{(1)}$ to $\text{EncDec}^{(32)}$. For $\text{EncDec}^{(1)}$, we use a normal random initialisation, and train for 3000 epochs until a stable error. For $\text{EncDec}^{(N)}$ with $N > 1$, we use the parameters trained from $\text{EncDec}^{(N-1)}$ as the initialisation. For the parameters $Q, P, b_1, b_2$, we pad them to match the size of $\text{EncDec}^{(N)}$ with normal distributions as initialisations.

It is shown that the low rank approximation phenomena discovered in the linear setting also appears in this nonlinear case.

