# OpenReview forum: "On the approximation properties of recurrent encoder-decoder architectures"
_ICLR.cc/2022/Conference — ICLR 2022 Spotlight_

### Official Review · Reviewer_WtE5 · 2021-11-02

**Correctness:** 4
**Technical Novelty And Significance:** 4
**Empirical Novelty And Significance:** Not applicable
**Recommendation:** 8
**Confidence:** 3

**Main Review:**

Overall, the paper is very well written. The summary of the theoretical results in the paper is neat, clear, and important. Detailed proof is provided for all propositions and theorems. The authors properly emphasize the difference between this work and previous related work.

My only concern is that I think it would be helpful for the readability by providing a brief introduction of the time-homogeneous functionals, Eq. 6 - 9, besides the math definition, though readers could find more detailed information in Li et al. (2021). Also, In Equation 5, h_\tau is not clearly defined as the last hidden state.

**Summary Of The Paper:**

This paper provides a theoretical analysis of both the universal approximation and the approximation rate of the RNN encoder-decoder in a linear setting.

**Summary Of The Review:**

This paper gives interesting notions and discoveries on the approximation properties of the recurrent encoder-decoder structure, and the statements are rigorous.

---

> ### Author Response · Authors · 2021-11-16
> **Authors' response**
>
> 1. ''My only concern is that I think it would be helpful for the readability by providing a brief introduction of the time-homogeneous functionals, Eq. 6 - 9, besides the math definition.''
>
>     - **We have added discussions of time-homogeneity in Remark 3.2, and included examples to illustrate its physical significance.**
>     Intuitively, time-homogeneity means that when the input is shifted, the output is also shifted accordingly in the same direction, by the same amount.
>     Convolution is a simple example to illustrate time-homogeneity.
>     One can regard the convolution as a smoothing operation, i.e. the output is a smoothed version of the input.
>     In this manner, when the input is shifted, the output should be shifted simultaneously.
>     As for examples of time-homogeneity in real-world applications, one can consider the audio-to-text conversion, since the timestamps for the input and output should be matched in this problem.
>     As a comparison, the video captioning task is an example of time-inhomogeneity.
>     When a sequence of frames is given, we would like to summarise the content of videos into sentences. In this case, the indices of outputs may have little relation with the timestamps of inputs, hence the output may not shift according to the input.
>
>
>
> 2. **Minor issues are fixed.**

---

### Official Review · Reviewer_Qn54 · 2021-11-02

**Correctness:** 3
**Technical Novelty And Significance:** 3
**Empirical Novelty And Significance:** 3
**Recommendation:** 8
**Confidence:** 3

**Main Review:**

I'd like to thanks the authors for their submission. I enjoyed reading it.

Overall, I think the paper is well-written, well-motivated, and is a good theoretical insight into one of the first types of DNN structures for sequence to sequence translation. I appreciate the fact that in a few instances (e.g. "The rank concept of temporal relationships") the paper poses a question, takes its time to explain the concepts and its generalization while trying to build intuition by explaining it in lower dimensional examples. It also does a good job of distinguishing itself from a previous closely related paper, Li et al. (2021). However, I have to say that since I am not following this line of research closely I cannot judge this work in context of existing works.

There are, however, a few concerns and questions:
1.  Authors mention "classical" or "traditional" RNN a few times. It is a bit vague to me what the opposite of that refers to? The RNN that the paper introduces here?
2.  In the related work section, it would be interesting to see what your type of analysis enables us in compare to other works.
3.  Eq. 1 and 2: any intuition as to why input space should vanish at infinity and output space should be bounded?
4.  Before eq. 4: "resulting from convolve" $\rightarrow$ "resulting from convolving"?
5.  Eq. 5 and 6 and the surrounding text is confusing. 5.1. $\sigma_E, \sigma_D, \sigma_O, \tau$ are not defined. 5.2. coding vector is introduced as $c$ which doesn't show up in equations. Again, coding vector is named $v$. 5.3. The equation for $g_t$ seems to be wrong? From reading the text I think it should be $g_t=\sigma_D(W_D g_{t-1} +b_D)$. 5.4. $h_0$ is missing.
6.  Is there any ramification for "linear and continuous-time idealization" before eq 6? A more intuitive explanation of this assumption is also appreciated.
7.  Eq. 7 and 8: the fact that one needs negative real parts in eigenvalues to ensure stability over long horizons is not explained.
8.  Authors on occasions focus on time-homogeneity and use it to distinguish their work from Li et al. (2021). In my opinion, it would be better if authors introduce this concept, its significance and motivation a bit sooner in the paper.
9.  Second paragraph in section 4.2: "where the dimension of the $N$..." $\rightarrow$ "where $N$..."
10.  Eq 11: $C^{(\alpha+1)}$ with superscript is not defined.
11.  Theorems 4.2 and 4.3 both discuss existence but not construction of these approximations. There is a section in appendix that discusses numerical illustrations. A discussion on algorithms to find these approximations and their nuances, even a short one, in the main text is appreciated.
12.  A discussion of the type of important assumptions that are used make these theoretical insights, how strong they are, and how far they are from the architectures that practitioners use today is welcomed.

**Summary Of The Paper:**

This paper provides theoretical insight for approximation properties of RNN encoder-decoder architecture in linear setting. More specifically, they study supervised learning problem of temporal modelling where a first RNN encodes a given sequence into a coding vector and a second RNN is responsible to decode said vector to a target output sequence. Linear setting here refers to a linear and continuous-time idealization in eq 6.

Their analysis summarized as following:
1.  Universal approximation: any linear, continuous, and regular temporal relation can be approximated by RNN encoder-decoder up to arbitrary accuracy.
2.  Approximation rate for large size coding vector: beside width of encoder and decoder, this error bound depends on $\alpha$ smoothness of $\mathbf{H}$ and $\beta$ temporal decay rates of the output of constant signal under $\mathbf{H}$ where $\mathbf{H}$ is a sequence of linear, continuous, and regular functionals on inputs. Each $\mathbf{H}$ has a unique two-parameter representation $\rho(t,s)$. Target functionals that are smooth and have fast decaying memory are identified as good target of this approximation.
3.  Approximation rate for small size coding vector: the architecture and the following assumptions in the paper give rise to an intrinsic structure to this type of RNN encoder-decoders called temporal product structure which can be deconstructed into encoder and decoder parts. By relating this structure to rank of temporal relationships they show approximation rate (in small size coding vector) is additionally a function of rank structure of target relationship. This analysis enables us to see the coding vector size as a knob to control number of parameters vs approximation error.
4.  Experiments: they show that the theoretical analysis holds in their experiments.

**Summary Of The Review:**

Generally a good paper that tries to theoretically explain the approximation power of RNN encoder-decoder architecture, although in linear and continuous-time setting. It provides intuition for the required characteristics of the target to ensure low approximation error and how to control number of parameters vs approximation error. A few places in the text and equations need authors' attention and better explanation of some assumptions are required.

---

> ### Author Response · Authors · 2021-11-16
> **Authors' response**
>
> 4. ''Eq. 7 and 8: the fact that one needs negative real parts in eigenvalues to ensure stability over long horizons is not explained.''
>
>     - Since the model has a closed form with the representation $c^\top e^{Vt} P Qe^{W s} U$ (see Eq. 9), which is a matrix exponential, its asymptotic scale as $t$ and $s$ goes to infinity is determined by real parts of the corresponding eigenvalues. If any of them is positive, the output will blow up exponentially.
>     If there are eigenvalues with 0 real-parts, then the functional will not be continuous (constant input signal may give unbounded output signal).
>     To avoid this, we assume the stability conditions, i.e. all the eigenvalues have negative real parts.  **We have added the above explanation in the main text (after Eq. 9).**
>
>
>
> 5. ''Authors on occasions focus on time-homogeneity and use it to distinguish their work from Li et al. (2021). In my opinion, it would be better if authors introduce this concept, its significance and motivation a bit sooner in the paper.''
>
>     - **We have added the discussion of time-homogeneity in Remark 3.2, and also provided examples to motivate it**.
>     We refer to the reply to **Reviewer WtE5** for detailed discussions.
>
>
>
> 6. ''Theorems 4.2 and 4.3 both discuss existence but not construction of these approximations. There is a section in appendix that discusses numerical illustrations. A discussion on algorithms to find these approximations and their nuances, even a short one, in the main text is appreciated.''
>
>     -  Our approximation results reported in the present paper guarantee the existence of efficient approximations.
>     In practice, it is not always possible to find such approximations.
>     We may use some optimisation/training methods to obtain a local optimal solution, which is often an upper bound of the approximation error.
>     As for specific algorithms, the approach in our numerical illustrations (details described in Appendix E.2) is one of them.
>     However, a rigorous evaluation of the training error and its relation to approximation concern the optimisation aspect, which is beyond the scope of current work. **We have added the above discussion in the Numerical illustrations part.**
>
>
>
> 7. ''A discussion of the type of important assumptions that are used make these theoretical insights, how strong they are, and how far they are from the architectures that practitioners use today is welcomed.''
>
>     We make a point-by-point discussion on the assumptions adopted in this paper.
>
>     - Continuous-time: the continuous-time assumption is not strong, as we
>     discussed in the reply to **Reviewer e39c**, point 1 and **Reviewer Qn54**, point 3.
>     Here, we summarise some of the important points related to practitioners.
>     For many problems in physics and engineering, if one can assume that the data is sampled (even irregularly) from some continuous dynamics, our analysis can be applied.
>     If not, numerical methods such as interpolation can be applied to convert the problem to a continuous-time case.
>
>     - Smoothness: it is just a technical assumption following the continuous-time assumption. If real data is continuous, it is often smooth or has smooth representation. Otherwise, a smooth interpolation can be applied.
>
>     - Decay of sequence length: this is not a strong assumption, as practical time series are always finite and can be padded with zeros to ensure decay.
>
>     - Decay of memory of target relationship: in this work (and also [1] for RNNs), we show that memory decay is a crucial property determining approximation capabilities. In real applications, such memory structures may or may not be easy to measure, and there are certain applications (periodic relationships) that do not decay at all. In this case, our results suggest that approximating them with stable RNN encoder-decoder structures may not be appropriate.
>
>     - Linearity: this assumption has been discussed in the reply to **Reviewer e39c**, Point 1 and **Reviewer Qn54**, Point 3.
>
>
>
> 8. Typos and notation issues are fixed and clarified.
> In particular, we would like to clarify the name "classical/traditional  RNN" used in the original manuscript.
> It refers to the Elman structure of recurrent neural networks [2] that maps sequences into sequences. To avoid confusion, we change the name to "RNNs".
> We have modified this in the revised version.
>
> References:
>
> [1] Zhong Li, Jiequn Han, Weinan E, and Qianxiao Li.  On the curse of memory in recurrent neural networks:  approximation and optimization analysis.  International Conference on Learning Representations, 2021.
>
> [2] Jeffrey L Elman. Finding structure in time. Cognitive science, 14(2):179–211, 1990

---

> ### Author Response · Authors · 2021-11-16
> **Authors' response**
>
> 1. ''In the related work section, it would be interesting to see what your type of analysis enables us in compare to other works.''
>
>     -  In this paper, we utilise the framework of functional analysis, which is previously used in Li et al. (2021) and Jiang et al. (2021) to analyse other time-series analysis architectures (please refer to [1,2] in the references of the reply to **Reviewer e39c**). Under this framework, the target temporal relationship can be formulated as a family of functionals. Furthermore, given a specific model/architecture, one can characterise targets that can be efficiently approximated based on properties induced by the corresponding model. For instance, Li et al. (2021) proves that RNNs are good at approximating smooth targets possessing fast decayed memories, and Jiang et al.(2021) shows that dilated CNNs can efficiently approximate targets with certain spectrum regularities.
>     In the present work, we found that the performance of recurrent encoder-decoders is related with memory, as well as certain rank structures.
>     This is a new property unique to encoder-decoder structures.
>     Therefore, the functional approximation framework considered here  enables detailed and precise comparisons between different temporal architectures. **In the revision, we have added discussions in the related work section, and provided comparisons where-ever possible with previous theoretical work in the main text.**
>
>
> 2. ''Eq. 1 and 2: any intuition as to why input space should vanish at infinity and output space should be bounded?''
>
>     - Input space: we assume the input vanishing at infinity for the following reasons.
>         - This requirement is a regularity condition, and it is mathematically convenient to work with such spaces in the functional analysis setting.
>         Specifically, we utilise the Riesz representation to derive main theorems, which requires the input time-series to vanish at infinity.
>         - The linearity of the RNN hypothesis space (as discussed in Proposition  3.1) requires $h_{-\infty}=0$
>         as the initial hidden state. This can be guaranteed by $x_{-\infty}=0$. In fact, we have
>         $$\frac{d}{ds} h_s = Wh_s+Ux_s,~h_{-\infty}=0
>         \Rightarrow
>         h_s=\int_{0}^{\infty} e^{W r} U x_{s-r} dr.$$
>         Then the above argument follows.
>         - The assumption also holds in reality. It is often the case that in applications, the inputs are finite sequences (possibly with arbitrary lengths), certainly vanishing at infinity.
>
>      - Output space: here, we consider linear continuous functionals (necessarily bounded), and continuous inputs vanishing at infinity (hence bounded), therefore the outputs are also bounded.
>
>
>
> 3. ''Is there any ramification for "linear and continuous-time idealization" before Eq. 6? A more intuitive explanation of this assumption is also appreciated. ''
>
>     - Linearity: first, the linear assumption on the model actually enables us to solve for a closed form of the model dynamics, which allows us to derive various theories together with the linear assumption on the target relationships. In addition, previous works on different architectures (Li et al. 2021, Jiang et al. 2021) also made a similar linear assumption on the model.
>     Thus, a natural comparison can be performed on precise similarities and differences of model forms and properties.
>     Finally, the insights from the linear setting can be  generalised.
>     As is shown in the added experiment (**Figure 3**), the key insight of low-rank approximation also holds in a tested nonlinear case.
>     We refer to the reply to **Reviewer e39c**, Point 1 for a more detailed discussion.
>
>     - Continuous-time: first, compared to the discrete case, the continuous-time setting makes it possible to use many powerful tools for mathematical analysis, such as Jackson approximation theorem.
>     In addition, the two settings can be related through methods in numerical analysis. Last, all the experiments are performed on discretised time-series data but show consistent results to the theoretical analysis.
>     We again refer to the reply to **Reviewer e39c**, Point 1 for a more detailed discussion.

---

### Official Review · Reviewer_e39c · 2021-11-03

**Correctness:** 3
**Technical Novelty And Significance:** 2
**Empirical Novelty And Significance:** 2
**Recommendation:** 6
**Confidence:** 2

**Main Review:**

This paper studies the approximation properties of RNN encoder-decoders, in the perspective of linear and continuous time. The authors first show the universal approximation property of a linear RNN encoder-decoder, then extends to the approximation rates that can be characterized by the rank of temporal products. The authors show that the temporal product structure may be the intrinsic structure arising from the encoder-decoder architecture.

Pros:
* Rigorous mathematical proofs that the encoder-decoder using linear RNN as the base building block is a more general form of sequence model that includes linear RNNs in its hypothesis space.
* Mathematically prove that the encoder-decoder architectures are a better fit for modeling time-inhomogenous sequence modeling problems.

Cons:
* The impact is somewhat limited by assuming linear and continuous-time for proving the approximation properties of RNN encoder-decoder, which is not the case in reality.
* Requires high-level mathematical skills to follow the paper, which might not be a good fit for ICLR considering the main audience who usually seeks rigorous experiments that back up the claims.

**Summary Of The Paper:**

This paper provides theoretical studies for why encoder-decoder can be seen as a generalization of RNNs in time-inhomogenous sequence modeling. The authors put an impressive amount of effort into mathematically defining approximation properties of RNN encoder-decoders beginning from a universal approximation result.

The authors first show the universal approximation property of RNN encoder-decoders, and subsequently, they show approximation rates of targets for RNN encoder-decoders. They introduce a notion of temporal products that can characterize the temporal relationships in the input/output pair.

**Summary Of The Review:**

Overall, a good mathematical study is important to have in a fast-moving field that certainly lacks theoretical results. However, I find that this study might not be that helpful practically as the setting is limited in linear RNN encoder-decoder setup, which is unlikely to be considered as a main choice of architecture these days. Also, as an ML researcher, it is a bit hard to follow a paper that is written in a formal mathematical style with a very small experiment at the end.

---

> ### Author Response · Authors · 2021-11-16
> **Authors' response**
>
> 2. ''Requires high-level mathematical skills to follow the paper, which might not be a good fit for ICLR considering the main audience who usually seeks rigorous experiments that back up the claims.''
>     - First, as a theoretical paper, being mathematically rigorous is one of our goals. We aim to precisely study the approximation properties of encoder-decoder architectures, and reveal how it fundamentally differs from other competitors in temporal modelling. Thus, our analysis must be presented rigorously for the theoretical community. In addition, our results reveal the functionality of the coding vector, and set an initial step towards understanding similarities and differences between different architectures, which are important insights and also useful for the practical community.
>     Finally, providing detailed practical guidance such as an explicit model/hyper-parameter selection algorithm is certainly an ultimate goal, but this is beyond the scope of the present theoretical paper.
>
>
>
> References:
> [1] Zhong Li, Jiequn Han, Weinan E, and Qianxiao Li. On the curse of memory in recurrent neural networks: approximation and optimization analysis. In *International Conference on Learning Representations*, 2021.
> [2] Haotian Jiang, Zhong  Li, and  Qianxiao  Li. Approximation theory of convolutional architectures for time series modelling. In *International Conference on Machine Learning*, 2021.
> [3] Edward N. Lorenz. Predictability: A problem partly solved. In *Proc. Seminar on predictability*, volume 1, 1996.

---

> > ### Comment · Reviewer_e39c · 2021-11-22
> > **Thanks for the rebuttal.**
> >
> > I read the rebuttal and the updated manuscript, thanks for addressing my comments! And I agree that theoretical support is important as the field grows. The authors kindly added more experiments and clarified the points that I was confused about. I agree that for theoretic community, being mathematically rigorous is the first priority, but also to make sure the paper has more broader impact, making it more easily readable for broader range of the audience is still valuable. I believe there will be a good middle ground for that. The authors already kindly addressed my concern about readability for the broader audience. I increased my review score by 1 point, and aligned my review with other reviewers towards acceptance.

---

> ### Author Response · Authors · 2021-11-16
> **Authors' response**
>
> 1. ''The impact is somewhat limited by assuming linear and continuous-time for proving the approximation properties of RNN encoder-decoder, which is not the case in reality.''
>
>     - Linearity assumption: first, we would point out that the linear setting is regarding the inputs instead of time.
>     On the side of target relationships (ground truths), linearity also appears in practical applications.
>     For instance, in signal processing, the widely-applied Fourier/Laplace/Wavelet transformations are all linear with respect to inputs.
>     In addition, on the side of models (hypothesis spaces), a rigorous study of the recurrent encoder-decoder architecture, even with linear activations, is still lacking to the best of our knowledge. This work sets a starting point for the possible theoretical analysis of general cases in the future.
>     Furthermore, in the present work, we rigorously treat the linear setting in a mathematical way. The results here reveal precisely how the coding vector works to trade off between the approximation error and model complexity, which is the key property of the recurrent encoder-decoder architecture.
>     Moreover, the linear setting was also considered in prior work on time-series analysis of different architectural classes (e.g. [1,2]). By adopting the same theoretical set up, we can perform immediate comparisons between different architectures (encoder-decoder, RNN, CNN) in temporal modelling in a mathematically precise manner.
>
>
>         **Finally, to demonstrate that the insights of our results (e.g. low rank approximation) carry over to nonlinear cases, we have added experiments (see Figure 3). Here, a nonlinear target temporal relationship (Lorenz 96 system [3]) and RNN encoder-decoder model (with nonlinear tanh activations) are numerically tested.**
>     The forced Lorenz 96 system is a high-dimensional nonlinear dynamical system with memory effects, with parameters $K, J$  controlling the overall complexity (see Appendix E.3).
>     We use nonlinear encoder-decoder models to learn the forced Lorenz system with a variety of coding vector sizes.
>     From Figure 3, we observe that the error decrements saturate when increasing the coding vector size $N$ beyond a threshold.
>     Further, this threshold increases with increasing target complexity $K$.
>     This is consistent with the analysis from the linear case, where we showed that increasing $N$ beyond the rank of the target relationship does not give better approximation bounds.
>     The additional nonlinear experiments here show that an effective version of this rank also exists in the Lorenz systems that affect approximation in the same way.
>     However, a precise definition of rank and theoretical approximation analysis in the nonlinear regime is beyond the current scope.
>
>
>
>     - Continuous-time assumption: first, the continuous-time setting enables us to utilise many mathematical tools (e.g. ODE solution methods, Jackson approximation theorem) to give a complete analysis and characterisation of the temporal modelling problem.
>     In addition, the continuous-time results can be connected to corresponding ones in discrete-time case through numerical analysis (e.g. finite-difference discretisation).
>     Furthermore, our experiments are performed in the discretized setting, and the numerical results obtained are consistent with theoretical analysis.
>     Finally, in real applications such as physics and engineering, it is often the case that the data is sampled from a (hidden) dynamical system, which is continuous in time.
>     Another case that necessitates continuous-time treatment is irregularly sampled time-series data (at different time length intervals).
>     Therefore, we think that analysing under the continuous-time still has a wide range of applications.

---

### Author Response · Authors · 2021-11-16
**To all reviewers.**

1. We would like to thank all the reviewers for their comments and suggestions to improve this paper.

2. We have answered every concern in the comments below.
In addition, based on the feedback, we have revised our paper and highlighted major changes in blue.
The corresponding changes are emphasised in boldface in the response.

3. Main concerns about the paper are that it may be too mathematical and results may be limited by linearity. We emphasise that our goal is to study the encoder-decoder architecture rigorously, and theoretical analysis is also an important part of the ML community.
Besides the mathematical theories, we have also included intuitions and motivations about the theoretical results, which can help practitioners to obtain better understandings.
We have improved this paper in this regard, according to comments raised by reviewer e39c.

4. **We have also added experiments (Figure 3) in the revised version.**
The experiments take the forced Lorenz 96 system as the target temporal relationship, which is nonlinear and high-dimensional.
This is an important example of reduced order modelling for convection dynamics, with applications in weather forecasting.
Note that the forced Lorenz system parameterise a highly nonlinear functional.
In addition,  we also added nonlinearity to the RNN encoder-decoder model by including the tanh activation.
The numerical results show that although the present theories are developed under the linear regime, the key insight of low-rank approximation also holds in this nonlinear case.
Thus, we believe that our results can contribute to practical understandings of encoder-decoder structures beyond the idealised theoretical setting considered here.

---

### Decision · Program_Chairs · 2022-01-20

**Decision:**

Accept (Spotlight)

**Comment:**

This paper presents a theoretical analysis of the approximation properties of linear recurrent encoder-decoder architectures, obtaining universal approximation results and subsequently showing approximation rates of targets for RNN encoder-decoders. It introduces a notion of "temporal products," which helps to characterize the types of temporal relationships that can be efficiently learned in this setting.

Overall, the reviewers and I all agree that this paper makes important theoretical contributions to the important problem of the approximation capabilities of encoder-decoder architectures. The main weaknesses involve the rather simplified linear problem setup, but this limitation is easily forgiven in this first-of-its-kind rigorous analysis. I recommend acceptance.